# LEARNING UNDER TEMPORAL LABEL NOISE

**Sujay Nagaraj**[1]    **Walter Gerych**[2]    **Sana Tonekaboni**[3]    **Anna Goldenberg**[1]
**Berk Ustun**[4]    **Thomas Hartvigsen**[5]
[1]University of Toronto    [2]MIT    [3]Broad Institute    [4]UCSD    [5]University of Virginia
Corresponding authors: s.nagaraj@mail.utoronto.ca, hartvigsen@virginia.edu

## ABSTRACT

Many time series classification tasks, where labels vary over time, are affected by *label noise* that also varies over time. Such noise can cause label quality to improve, worsen, or periodically change over time. We first propose and formalize *temporal label noise*, an unstudied problem for sequential classification of time series. In this setting, multiple labels are recorded over time while being corrupted by a time-dependent noise function. We first demonstrate the importance of modeling the temporal nature of the label noise function and how existing methods will consistently underperform. We then propose methods to train noise-tolerant classifiers by estimating the temporal label noise function directly from data. We show that our methods lead to state-of-the-art performance under diverse types of temporal label noise on real-world datasets.[1]

## 1    INTRODUCTION

Supervised learning datasets often contain incorrect, or *noisy*, observations of ground truth labels. Such *label noise* commonly arises during human annotation [1, 28] when labelers lack expertise [31, 77], labeling is hard [13, 30, 77], labels are subjective [49, 57, 62], or during automatic annotation where there are systematic issues like measurement error [32, 52]. These factors, among others, make label noise prevalent in supervised learning. Label noise is a key vulnerability in supervised learning [17, 25, 75]. Intuitively, models trained with noisy labels may learn to predict noise – such models then underperform at test time when attempting to predict the ground truth [26, 42].

Label noise has been studied extensively, producing a stream of works on *static* prediction tasks where we predict single outcomes [3, 43, 45, 67]. Many *non-static* prediction tasks (i.e., time series) have features and labels that evolve over time. In these cases, it is only natural for label noise to also evolve over time – label noise can be *temporal*. This notion of time-varying noise has yet to be formalized, yet it is found in numerous real-world tasks. For example:

- *Human Activity Recognition*. Wearable device studies often ask participants to annotate their activities (e.g., exercise) over time. Participants often mislabel their activities due to recall bias, time of day, or labeling-at-random for monetized studies [27, 61].

- *Self-Reported Outcomes for Mental Health*: Mental health studies often collect self-reported survey data (e.g., depression) over long periods of time. Such self-reporting is known to be biased [5, 53, 55, 72, 46], as participants are more or less likely to report certain outcomes. For example, the accuracy of self-reported alcohol consumption is often seasonal [8].

- *Clinical Measurement Error*: Clinical prediction models often predict outcomes (e.g., mortality) derived from clinician notes in electronic health records. These labels may capture noisier annotations during busier times – e.g., when a patient is deteriorating [76].

Addressing the problem of temporal label noise is challenging from a technical perspective. Existing methods for learning under static label noise [48, 50] are not flexible enough to account for noise rates varying over time – these approaches have no mechanism to capture temporal label noise and therefore implicitly misspecify the noise model. Additionally, the label noise function, temporal or otherwise, is often unknown; most datasets lack an indication for which instances or time steps are

---

[1]All of our code is available at https://github.com/sujaynagaraj/TemporalLabelNoise

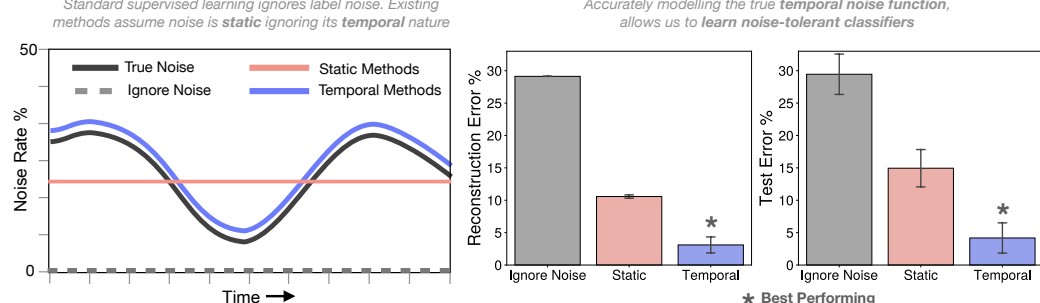

**Figure 1:** In time series tasks, label quality can vary over time due to *temporal* label noise. Existing methods assume noise can perform poorly as they assume a static noise model over time. We propose that accurate modeling of temporal label noise can improve performance. We demonstrate this by showing performance improvements on an activity recognition task where we compare reconstruction error and accuracy between static and temporal methods subject to 30% temporal label noise across 10 draws (see results for `moving` dataset in Appendix D.4 for details).

more likely to be accurate. Another practical limitation is the lack of ground truth data in real-world noisy datasets.

To address these challenges, we propose objective functions for sequence-to-sequence classification tasks in the context of time series data. These objectives *learn* the temporal noise function directly from time series data with temporally noisy labels. Our methods leverage a temporal loss function that is robust to temporal label noise. Our algorithms can be used *out of the box* and allow practitioners to learn noise-tolerant time series classifiers, even with unknown, temporal label noise. Our main contributions are:

1. We formalize the problem of learning from noisy labels in temporal settings.

2. We propose a novel loss function for training models that are robust to temporal label noise. On its own, we show this can already be used to improve prior methods.

3. We develop a suite of methods to learn time series classifiers under temporal label noise. Our methods can overcome the practical challenges of this regime – i.e., by estimating an unknown noise function from time series data.

4. We show the consequences of *not* accounting for temporal label noise by simply using existing methods for noisy labels – these methods perform poorly under temporal noise. Our results highlight the necessity of accounting for temporal noise, as our methods lead to more noise-robust classifiers – temporal or otherwise.

## 2 RELATED WORK

Our work is motivated by a series of problems that arise in machine learning in healthcare [20, 40, 46]. Existing approaches primarily rely on supervised learning to model changes in latent states over time [14, 35, 64, 68, 73, 85]. In practice, these underlying latent states can reflect clinical states that patients can transition in and out of (e.g., sick vs healthy). Existing approaches for this problem typically rely on modeling sequences with deep neural networks [11, 39, 65] and using attention-based mechanisms to prioritize relevant segments of data [71]. An understudied problem in this domain is label noise – what happens when inaccurate labels arise in real-world time series datasets?

The vast majority of work on noisy labels studies label noise in static prediction tasks where we predict a single outcome and where label noise cannot change over time [3, 43, 45, 67, 47] We consider how label noise can arise in time series – a prediction task where the noise rates can change over time. Recent work in this setting focuses on identifying noisy instances in time series, specifically by exploiting the notion that labels at neighboring time steps are unlikely to be corrupted together [see e.g., 2, 12]. In contrast, we focus on developing algorithms to train models that are robust to temporal label noise. With respect to existing work, our approach can account for label noise in tasks where labels at neighboring time steps are *more* likely to be corrupted together (e.g., due to

seasonal fluctuations in annotator error). We develop methods for empirical risk minimization with *noise-robust loss functions* [c.f., 21, 38, 43, 48, 74]. Our work establishes the potential to learn from noisy labels in this setting when we have knowledge of the underlying noise process [c.f., 48, 50], and when we are able to learn the noise model from noisy data [see e.g., 37, 38, 50, 79, 82, 86].

## 3 FRAMEWORK

**Preliminaries** We consider a temporal classification task over $C$ classes and $T$ time steps. Each instance is characterized by a triplet of *sequences* over $T$ time steps $(\mathbf{x}_{1:T}, \mathbf{y}_{1:T}, \tilde{\mathbf{y}}_{1:T}) \in \mathcal{X} \times \mathcal{Y} \times \mathcal{Y}$ where $\mathcal{X} \subseteq \mathbb{R}^{d \times T}$ represents a multivariate time-series and $\mathcal{Y} = \{1, \ldots, C\}^T$ represents a sequence of labels. Here, $\mathbf{x}_{1:T}$, $\mathbf{y}_{1:T}$, and $\tilde{\mathbf{y}}_{1:T}$ are sequences of instances, *clean labels* and *noisy labels*, respectively. For example, we can capture settings where $\mathbf{x}_{1:T}$, $\mathbf{y}_{1:T}$ are recordings from an accelerometer with true activity labels, and $\tilde{\mathbf{y}}_{1:T}$ represent noisy annotations of activity.

Under temporal label noise, the *true* label sequence $\mathbf{y}_{1:T}$ is unobserved, and we only have access to a set of $n$ noisy instances $D = \{(\boldsymbol{x}_{1:T}, \tilde{\boldsymbol{y}}_{1:T})_i\}_{i=1}^n$. We assume that each sequence in $D$ is generated i.i.d. from a joint distribution $P_{\mathbf{x}_{1:T}, \mathbf{y}_{1:T}, \tilde{\mathbf{y}}_{1:T}}$, where the label noise process can vary over time. This distribution obeys two standard assumptions in temporal modeling and label noise [4, 10, 69]:

**Assumption 1** (Future Independence). *A label at time $t$ depends only on the past sequence of feature vectors up to $t$:* $p(\mathbf{y}_{1:T} \mid \mathbf{x}_{1:T}) = \prod_{t=1}^T p(\mathbf{y}_t|\mathbf{x}_{1:t})$

**Assumption 2** (Feature Independence). *The sequence of noisy labels is conditionally independent of the features given the true labels:* $\tilde{\mathbf{y}}_{1:t} \perp\!\!\!\perp \mathbf{x}_{1:t} \mid \mathbf{y}_{1:t}$ *for* $t = 1, \ldots, T$

These assumptions are relatively straightforward, as they require that the current observation is independent of future observations and assume a feature-independent noise regime. Assumption 1 allows the joint sequence distribution to factorize as: $p(\tilde{\mathbf{y}}_{1:T}|\mathbf{x}_{1:T}) = \prod_{t=1}^T q_t(\tilde{\mathbf{y}}_t|\mathbf{x}_{1:t})$. Here, we introduce $q_t$, a slight abuse of notation, to denote a probability that is time-dependent. Assumption 2 allows for the noisy label distribution at $t$ to be further decomposed as:

$$q_t(\tilde{\mathbf{y}}_t|\mathbf{x}_{1:t}) = \sum_{y \in \mathcal{Y}} q_t(\tilde{\mathbf{y}}_t \mid \mathbf{y}_t = y) p(\mathbf{y}_t = y \mid \mathbf{x}_{1:t}) \tag{1}$$

### 3.1 LEARNING FROM TEMPORAL LABEL NOISE

Our goal is to learn a temporal classification model $\boldsymbol{h}_\theta : \mathbb{R}^{d \times t} \to \mathbb{R}^C$ with model parameters $\theta \in \Theta$ and $t \leq T$. Here, $\boldsymbol{h}_\theta(\mathbf{x}_{1:t})$ returns an estimate of $p(\mathbf{y}_t|\mathbf{x}_{1:t})$. To infer the label at time step $t$, $\boldsymbol{h}_\theta$ takes as input a sequence of feature vectors up to $t$, and outputs a sequence of labels by taking the $\arg\max$ of the predicted distribution for each time step (see e.g., [4, 69]). We estimate parameters $\hat{\theta}$ for a model robust to noise, by maximizing the expected accuracy as measured in terms of the *clean* labels:

$$\hat{\theta} = \underset{\theta \in \Theta}{\operatorname{argmax}} \, \mathbb{E}_{\mathbf{y}_{1:T}|\mathbf{x}_{1:T}} \prod_{t=1}^T p(\mathbf{y}_t = \boldsymbol{h}_\theta(\mathbf{x}_{1:t}) \mid \mathbf{x}_{1:t})$$

However, during training time we only have access to sequences of *noisy* labels. To demonstrate how we can sidestep this limitation, we first need to introduce a flexible way to model noise rates varying over time. Existing methods assume that noise is time-invariant (Fig. 1). To relax this assumption, we capture the temporal nature of noisy labels using a *temporal label noise function*, in Def. 1.

**Definition 1.** Given a temporal classification task with $C$ classes and noisy labels, the *temporal label noise function* is a matrix-valued function $\boldsymbol{Q} : \mathbb{R}_+ \to [0, 1]^{C \times C}$ that specifies the label noise distribution at any time $t > 0$.

We denote the output of the temporal noise function at time $t$ as $\boldsymbol{Q}_t := \boldsymbol{Q}(t)$. This is a $C \times C$ matrix whose $i, j^{\text{th}}$ entry encodes the flipping probability of observing a noisy label $j$ given clean label $i$ at time $t$: $q_t(\tilde{\mathbf{y}}_t = j \mid \mathbf{y}_t = i)$. We observe that $\boldsymbol{Q}_t$ is positive, row-stochastic, and diagonally dominant – ensuring that $\boldsymbol{Q}_t$ encodes a valid probability distribution [38, 50].

$\boldsymbol{Q}_\omega$ denotes a temporal noise function parameterized by a function with parameters $\omega$. This parameterization can be constructed to encode essentially any temporal noise function. As shown in Table 1, we can capture a wide variety of temporal noise using this representation.

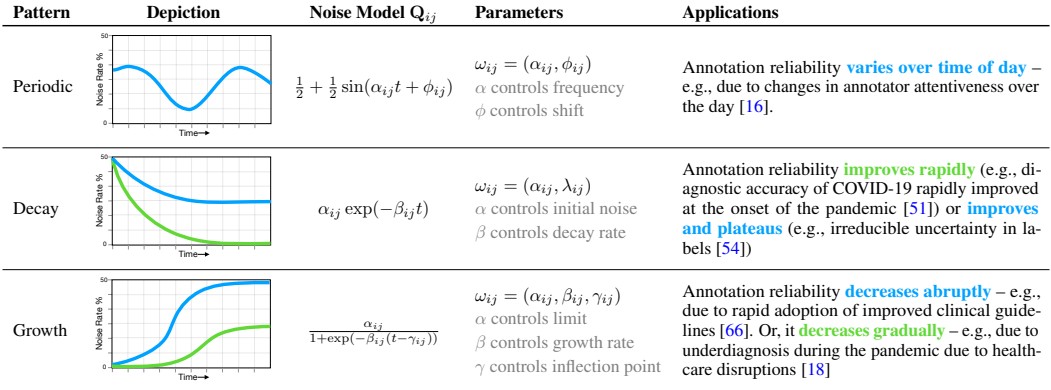

| Pattern | Depiction | Noise Model $Q_{ij}$ | Parameters | Applications |
|---|---|---|---|---|
| Periodic | | $\frac{1}{2} + \frac{1}{2}\sin(\alpha_{ij}t + \phi_{ij})$ | $\omega_{ij} = (\alpha_{ij}, \phi_{ij})$
$\alpha$ controls frequency
$\phi$ controls shift | Annotation reliability **varies over time of day** – e.g., due to changes in annotator attentiveness over the day [16]. |
| Decay | | $\alpha_{ij}\exp(-\beta_{ij}t)$ | $\omega_{ij} = (\alpha_{ij}, \lambda_{ij})$
$\alpha$ controls initial noise
$\beta$ controls decay rate | Annotation reliability **improves rapidly** (e.g., diagnostic accuracy of COVID-19 rapidly improved at the onset of the pandemic [51]) or **improves and plateaus** (e.g., irreducible uncertainty in labels [54]) |
| Growth | | $\frac{\alpha_{ij}}{1+\exp(-\beta_{ij}(t-\gamma_{ij}))}$ | $\omega_{ij} = (\alpha_{ij}, \beta_{ij}, \gamma_{ij})$
$\alpha$ controls limit
$\beta$ controls growth rate
$\gamma$ controls inflection point | Annotation reliability **decreases abruptly** – e.g., due to rapid adoption of improved clinical guidelines [66]. Or, it **decreases gradually** – e.g., due to underdiagnosis during the pandemic due to healthcare disruptions [18] |

**Table 1:** Overview of temporal noise functions for time series classification tasks (see Appendix D for other examples). We show the noise model $\boldsymbol{Q}(t)$ and parameters $\omega$ to model $q(\tilde{\mathrm{y}}_t = j \mid \mathrm{y}_t = i)$. Practitioners can account for temporal variation by choosing a parametric class to capture effects and fit parameters from data.

## 3.2 LOSS CORRECTION

Modeling temporal label noise is the first piece of the puzzle in training time series classifiers robust to temporal label noise. However, we still need to consider how to leverage these noise models during empirical risk minimization. It remains unclear if and how existing loss correction techniques work for time series. Here, we present theoretical results showing that learning is possible in our setting when we know the true temporal noise function $\boldsymbol{Q}(t)$. We include proofs in Appendix A.

We begin by treating the noisy posterior as the matrix-vector product of a noise transition matrix and a clean class posterior Eq. (1). To this effect, we define the *forward temporal loss*:

**Definition 2.** Given a temporal classification task over $T$ time steps, a noise function $\boldsymbol{Q}(t)$, and a proper composite loss function $\ell_t$ [58], the *forward temporal loss* of a model $\boldsymbol{h}_\theta$ on an instance $(\tilde{\boldsymbol{y}}_{1:T}, \boldsymbol{x}_{1:T})$ is:

$$\overrightarrow{\ell}_{seq}(\tilde{\boldsymbol{y}}_{1:T}, \boldsymbol{x}_{1:T}, \boldsymbol{h}_\theta) := \sum_{t=1}^{T} \ell_t(\tilde{y}_t, \boldsymbol{Q}_t^\top \boldsymbol{h}_\theta(\boldsymbol{x}_{1:t}))$$

An intriguing property of the *forward temporal loss* is that its minimizer over the noisy labels maximizes the likelihood of the data over the clean labels. This suggests that the *forward temporal loss* is robust to label noise:

**Proposition 1.** A classifier that minimizes the empirical *forward temporal loss* over the noisy labels maximizes the empirical likelihood of the data over the clean labels.

$$\underset{\theta \in \Theta}{\operatorname{argmin}}\, \mathbb{E}_{\tilde{\mathbf{y}}_{1:T}, \mathbf{x}_{1:T}}\, \overrightarrow{\ell}_{seq}(\tilde{\mathbf{y}}_{1:T}, \mathbf{x}_{1:T}, \boldsymbol{h}_\theta) = \underset{\theta \in \Theta}{\operatorname{argmin}} \sum_{t=1}^{T} \mathbb{E}_{\mathbf{y}_{1:t}, \mathbf{x}_{1:t}} \ell_t(\mathbf{y}_t, \boldsymbol{h}_\theta(\mathbf{x}_{1:t}))$$

Proposition 1 implies that we can *train on the noisy distribution* and learn a noise-tolerant classifier in expectation.

## 4 METHODOLOGY

Our results in the previous section show that we can train models that are robust to label noise. However, the temporal noise function $\boldsymbol{Q}$ is not available to the practitioner and must be learned from data. Our algorithm seeks to first learn this function from noisy data then account for noise using Def. 2. This strategy can be applied in multiple ways. In what follows, we propose a method where we jointly learn the temporal noise function and the model. We also present extensions for different use-cases in Section 4.3 and discuss how to choose between methods in Section 4.4

### 4.1 FORMULATION

We start with a method that simultaneously learns a time series classifier and temporal noise function. Given a noisy dataset, we learn these elements by solving the following optimization problem $\forall t \in [1, T]$:

$$
\begin{aligned}
\min_{\omega, \theta} \quad & \text{Vol}(\boldsymbol{Q}_\omega(t)) \\
\text{s.t.} \quad & \boldsymbol{Q}_\omega(t)^\top \boldsymbol{h}_\theta(\boldsymbol{x}_{1:t}) = p(\tilde{y}_t \mid \boldsymbol{x}_{1:t})
\end{aligned}
\tag{2}
$$

Eq. (2) is designed to return a faithful representation of the noise function $\hat{\omega}$ by imposing the *minimum-volume simplex* assumption [38], and a noise-tolerant temporal classifier $\hat{\theta}$ by minimizing the *forward temporal loss* in Def. 2.

Here, the objective minimizes the volume of the noise matrix, denoted as $\text{Vol}(\boldsymbol{Q}_t)$. This returns a matrix $\boldsymbol{Q}_t$, at each time step, that obeys the *minimum-volume simplex* assumption, which is a standard condition used to ensure identifiability of label noise in static tasks [see e.g., 38, 83]. In practice, this assumption ensures that $\boldsymbol{Q}_t$ encloses the noisy conditional data distribution at time $t$: $p(\tilde{y}_t \mid \mathbf{x}_{1:t})$. Containment ensures that the estimated noise matrix could have generated each point in the noisy dataset – i.e., so that the corresponding noisy probabilities $p(\tilde{y}_t \mid \mathbf{x}_{1:t})$ obey Eq. (1). Our use of this assumption guarantees the identifiability of $\boldsymbol{Q}_t$ when the posterior distribution is sufficiently-scattered over the unit simplex [see also 38, for details].

### 4.2 ESTIMATION PROCEDURE

The formulation above applies to any generic matrix-valued function according to Def. 1 with parameters $\omega$. Because time series classification tasks can admit many types of temporal label noise functions (Table 1), we must ensure $\omega$ has sufficient representational capacity to handle many noise functions. Therefore in practice, we instantiate our solution as a fully connected neural network with parameters $\omega$, $\boldsymbol{Q}_\omega(\cdot) : \mathbb{R} \to [0,1]^{C \times C}$, adjusted to meet Def. 1 (see Appendix D.3 for implementation details). We can now model any temporal label noise function owing to the universal approximation properties of neural networks.

Provided the function space $\Theta$ defines autoregressive models of the form $p(y_t | \boldsymbol{x}_{1:t})$ (e.g., RNNs, Transformers, etc.), we can solve Eq. (2) using an augmented Lagrangian method for equality-constrained optimization problems [6]:

$$
\mathcal{L}(\theta, \omega) = \frac{1}{T} \sum_{t=1}^{T} \left[ \|\boldsymbol{Q}_\omega(t)\|_F + \lambda R_t(\theta, \omega) + \frac{c}{2} |R_t(\theta, \omega)|^2 \right]
\tag{3}
$$

Here: $\|\boldsymbol{Q}_\omega(t)\|_F$ denotes the Frobenius norm of $\boldsymbol{Q}_\omega(t)$, which acts as a convex surrogate for $\text{Vol}(\boldsymbol{Q}_\omega(t))$ [7]. Likewise, $R_t(\theta, \omega) = \frac{1}{n} \sum_{i=1}^{n} \ell_t(\tilde{y}_{t,i}, \boldsymbol{Q}_\omega(t)^\top \boldsymbol{h}_\theta(\boldsymbol{x}_{1:t,i}))$ denotes the violation of the equality constraint for each $t = 1 \ldots T$. $\lambda \in \mathbb{R}_+$ is the Lagrange multiplier and $c > 0$ is a penalty parameter. Both these terms are initially set to a default value of 1, we gradually increase the penalty parameter until the constraint holds and $\lambda$ converges to the Lagrange multiplier for the optimization problem Eq. (2) [6]. This approach recovers the best-fit parameters to the optimization problem in Eq. (2). We call this approach *Continuous Estimation* and provide additional details on our implementation in Appendix B.

### 4.3 ALTERNATIVE APPROACHES

We describe two alternative approaches that extend existing label noise methods to temporal settings and may be valuable to use in tasks where we cannot assume continuity or have access to additional information.

**Discontinuous Estimation** Another approach is to assume that there is no temporal relationship between each $\boldsymbol{Q}_t$ across time and treat each time step independently. This approach is well suited for tasks where time steps are unevenly spaced (e.g., some clinical data involves labels collected over years or at different frequencies [56]). We can address such tasks through an estimation procedure where we learn a model that achieves the same objective without assuming continuity. In contrast to

Eq. (3), this approach $\hat{\boldsymbol{Q}}_t$ is parameterized with a *separate* set of trainable real-valued weights, fitting the parameters at each time step using the data from that time step. This approach can be generalized to extend any state-of-the-art technique for noise transition matrix estimation in the static setting (e.g., Li et al. [38], Yong et al. [83], etc.).

**Plug-In Estimation**   It is also advantageous to have a simple, plug-in estimator of the temporal noise function. Plug-in estimators are model-agnostic, can flexibly be deployed to other models, and can be efficient to estimate. We can construct a plug-in estimator of temporal label noise using *anchor points*, instances whose labels are known to be correct. Empirical estimates of the class probabilities of anchor points can be used to estimate the noise function. This estimate can then be plugged into Def. 2 to train a noise-tolerant time series classifier. Formally, in a temporal setting, anchor points [41, 50, 79] are instances that maximize the probability of belonging to class $i$ at time step $t$:

$$\bar{\boldsymbol{x}}_t^i = \arg\max_{\boldsymbol{x}_t} p(\tilde{\mathrm{y}}_t = i \mid \boldsymbol{x}_{1:t}) \tag{4}$$

Since $p(\mathrm{y}_t = i \mid \bar{\boldsymbol{x}}_{1:t}^i) \approx 1$ for the clean label, we can express each entry of the label noise matrix as:

$$\hat{\boldsymbol{Q}}(t)_{i,j} = p(\tilde{\mathrm{y}}_t = j \mid \bar{\boldsymbol{x}}_{1:t}^i) \tag{5}$$

We construct a plug-in estimate of $\hat{\boldsymbol{Q}}(t)$ using a two-step approach: we identify anchor points for each class and time step $t = 1, \ldots, T$ and replace each entry of $\hat{\boldsymbol{Q}}(t)$ with the empirical estimate defined in Eq. (5) This is a generalization of the approach of Patrini et al. [50] developed for static prediction tasks.

## 4.4 DISCUSSION

All three methods in Section 4.2 and 4.3 improve performance in temporal classification tasks by accounting for temporal label noise (see e.g., Fig. 1 and Section 5). However, they each have their strengths and limitations. Here we provide guidance for users to choose between methods:

- Continuous estimation imposes continuity across time steps – assuming that nearby points likely have similar noise levels i.e., $\hat{\boldsymbol{Q}}(t) \approx \hat{\boldsymbol{Q}}(t+1)$. This assumption can improve reconstruction in tasks with multiple time steps by reducing the effective number of model parameters. Conversely, it may also lead to misspecification in settings that exhibit discontinuity.
- Discontinuous estimation can handle discontinuous temporal noise processes – assuming the noise levels of nearby points are independent of one another. This approach requires fitting more parameters, which scales according to $T$. This can lead to computational challenges and overfitting, especially for long sequences.
- Plug-In estimation has a simpler optimization problem. This is useful in that we can use separate datasets to estimate the noise model and to train the classifier. In practice, the main challenges in this approach stem from verifying the existence of anchor points [79].

## 5 EXPERIMENTS

We benchmark our methods on a collection of temporal classification tasks from real-world applications. Our goal is to evaluate methods in terms of robustness to temporal label noise, and characterize when it is important to consider temporal variation in label noise. We include additional details on setup and results in Appendix D and code to reproduce our results on GitHub.

## 5.1 SETUP

We work with four real-world datasets from healthcare. Each dataset reflects binary classification tasks over a complex feature space, with labeled examples across multiple time steps, where the labels are likely to exhibit label noise. The tasks include:

1. `moving`: human activity recognition task where we detect movement states (e.g., walking vs. sitting) using temporal accelerometer data in adults [59].

2. `senior`: similar human activity recognition task as above but in senior citizens [44]

3. `sleeping`: sleep state detection (e.g., light sleep vs. REM) task using continuous EEG data [22]

4. `blinking`: eye movement (open vs. closed) detection task using continuous EEG data [60] from continuous EEG data.

We consider a semi-synthetic setup where we treat labels in the training sample as ground truth labels and create noisy labels by corrupting them with a noise function for different noise rates. This setup reflects a standard approach used to benchmark algorithms for learning under label noise [see e.g., 48, 50]. We consider six different label noise functions:

- *Linear*: noise rates linearly fall over time.
- *Decay*: noise rates decay exponentially over time.
- *Growth*: noise rates rapidly rise over time.
- *Periodic*: noise rates vary in a seasonal fashion.
- *Mixed*: noise rates for $y_t = 0$ grow and noise rates for $y_t = 1$ fall over time.
- *Static*: noise rates are constant over time.

We present two sample noise functions in Fig. 2, and a complete list in Fig. 4 in Appendix D.

We split each dataset into a *noisy* training sample (80%, used to train the models and correct for label noise) and a *clean* test sample (20%, used to compute unbiased estimates of out-of-sample performance).

We train time series classifiers on the aforementioned datasets using the three techniques described in Section 4: Continuous, Discontinuous, and Plug-In. We compare these models to baseline models that ignore label noise (Ignore) to loss-tolerant methods that assume a static noise model across time steps: Anchor [41, 50, 79] and VolMinNet [38].

We evaluate each model in terms of the following metrics:

- *Test Accuracy* measures how well each method performs on held-out test data with *clean* labels.
- *Approximation Error* characterizes how well each method learns the temporal noise function, using the $\mathrm{Error}(\boldsymbol{Q}_t, \hat{\boldsymbol{Q}}_t) := \frac{1}{T} \sum_{t=1}^{T} \|\boldsymbol{Q}_t - \hat{\boldsymbol{Q}}_t\|$ between the true $\boldsymbol{Q}_t$ and estimated $\hat{\boldsymbol{Q}}_t$ for all $t$.

## 5.2 RESULTS

We summarize our results for two temporal noise functions in Table 2 and Table 3. We evaluate the performance of methods across different noise models and attribute the gains to our ability to capture the underlying temporal noise function in Fig. 2. We include additional results for other noise models and for multi-class classification tasks in Appendix D.4. In what follows we discuss key results.

**On the Performance Gains of Modeling Temporal Noise** First, we show clear value in accounting for temporal label noise. Table 2 shows the performance of each method on all four datasets. We

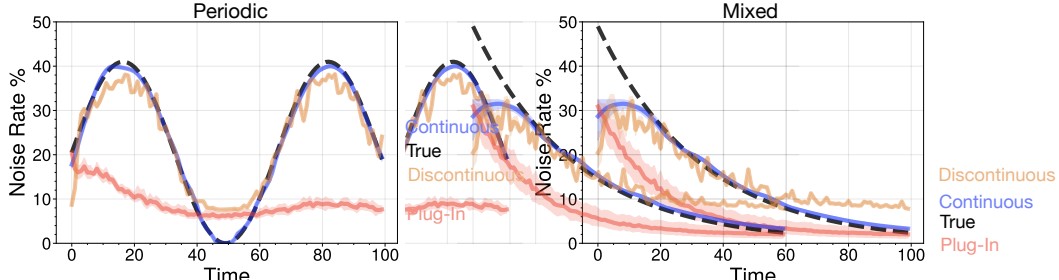

**Figure 2:** Comparison of the ground truth unseen temporal noise function $\boldsymbol{Q}(t)$ and its estimate $\hat{\boldsymbol{Q}}(t)$ from each *Temporal* method on the `moving` data. We show the noise rate for the negative class only for clarity. We show results for *Periodic* and *Mixed* noise regimes. As shown, our *Temporal* methods can learn the true label noise function across different noise patterns. The resulting noise models have lower reconstruction error and are superior to static approaches.

| Dataset | Metric | Ignore | Static | | Temporal | | |
| --- | --- | --- | --- | --- | --- | --- | --- |
| | | | Anchor | VolMinNet | Plug-In | Discontinuous | Continuous |
| moving [59] | Test Error | $29.4 \pm 1.7\%$ | $20.9 \pm 2.6\%$ | $14.9 \pm 2.7\%$ | $20.0 \pm 1.8\%$ | $15.9 \pm 2.7\%$ | **$4.2 \pm 2.2\%$** |
| $n = 192, d = 14, T = 50$ | Approx. Error | – | $42.4 \pm 3.4\%$ | $35.3 \pm 0.8\%$ | $36.8 \pm 1.7\%$ | $32.6 \pm 0.5\%$ | **$10.3 \pm 3.9\%$** |
| senior [44] | Test Error | $22.7 \pm 1.7\%$ | $20.7 \pm .01\%$ | $19.0 \pm 0.7\%$ | $18.8 \pm 1.1\%$ | $13.6 \pm 1.2\%$ | **$11.0 \pm 0.3\%$** |
| $n = 444, d = 6, T = 100$ | Approx. Error | – | $35.9 \pm 2.5\%$ | $36.3 \pm 0.4\%$ | $26.9 \pm 1.2\%$ | $21.7 \pm 0.2\%$ | **$6.4 \pm 0.8\%$** |
| blinking [60] | Test Error | $34.1 \pm 2.0\%$ | $34.1 \pm 2.3\%$ | $29.6 \pm 2.2\%$ | $29.6 \pm 2.8\%$ | $29.9 \pm 3.0\%$ | **$29.6 \pm 2.3\%$** |
| $n = 299, d = 14, T = 50$ | Approx. Error | – | $35.3 \pm 0.8\%$ | $35.2 \pm 0.7\%$ | $19.6 \pm 1.1\%$ | $26.6 \pm 0.9\%$ | **$14.9 \pm 2.3\%$** |
| sleeping [22] | Test Error | $28.7 \pm 0.8\%$ | $24.9 \pm 1.1\%$ | $26.8 \pm 1.4\%$ | $20.4 \pm 1.8\%$ | $19.6 \pm 0.8\%$ | **$16.3 \pm 0.4\%$** |
| $n = 964, d = 7, T = 100$ | Approx. Error | – | $34.3 \pm 1.8\%$ | $41.8 \pm 0.1\%$ | $19.1 \pm 3.5\%$ | $22.4 \pm 0.2\%$ | **$4.9 \pm 0.5\%$** |

**Table 2:** Model performance and approximation error for all methods and datasets when labels are corrupted using a *Mixed* noise function (average 30% label noise across all time steps, one class with decreasing noise rate, one class with increasing noise rate). We report the clean test error (%) and mean approximation error of $\hat{Q}(t)$ $\pm$ st.dev over 10 runs. The best-performing methods are highlighted in Green. Continuous outperforms all baselines.

| Dataset | Metric | Ignore | Static | | Temporal | | |
| --- | --- | --- | --- | --- | --- | --- | --- |
| | | | Anchor | VolMinNet | Plug-In | Discontinuous | Continuous |
| moving [59] | Test Error | $24.0 \pm 5.1\%$ | $18.0 \pm 3.6\%$ | $13.5 \pm 6.0\%$ | $18.5 \pm 4.3\%$ | $14.0 \pm 5.7\%$ | **$1.7 \pm 0.6\%$** |
| $n = 192, d = 14, T = 50$ | Approx. Error | – | $49.9 \pm 4.7\%$ | $43.5 \pm 3.0\%$ | $48.1 \pm 4.4$ | $33.5 \pm 5.0\%$ | **$9.1 \pm 1.6\%$** |
| senior [44] | Test Error | $22.5 \pm 2.2\%$ | $20.1 \pm 1.5\%$ | $16.4 \pm 2.9\%$ | $19.9 \pm 1.3\%$ | $14.1 \pm 2.6\%$ | **$10.6 \pm 0.7\%$** |
| $n = 444, d = 6, T = 100$ | Approx. Error | – | $47.3 \pm 4.0\%$ | $42.4 \pm 3.9\%$ | $46.9 \pm 3.9\%$ | $20.5 \pm 2.1\%$ | **$9.7 \pm 2.7\%$** |
| blinking [60] | Test Error | $37.4 \pm 3.1\%$ | $35.8 \pm 2.7\%$ | $33.2 \pm 2.8\%$ | $35.8 \pm 2.3\%$ | $32.0 \pm 2.9\%$ | **$30.4 \pm 3.1\%$** |
| $n = 299, d = 14, T = 50$ | Approx. Error | – | $49.3 \pm 4.2\%$ | $42.4 \pm 3.9\%$ | $46.4 \pm 4.4\%$ | $24.7 \pm 3.3\%$ | **$10.5 \pm 3.3\%$** |
| sleeping [22] | Test Error | $23.3 \pm 3.1\%$ | $22.8 \pm 2.7\%$ | $22.4 \pm 3.0\%$ | $22.1 \pm 2.0\%$ | $20.1 \pm 3.5\%$ | **$15.2 \pm 0.8\%$** |
| $n = 964, d = 7, T = 100$ | Approx. Error | – | $44.9 \pm 4.0\%$ | $42.6 \pm 4.1\%$ | $43.9 \pm 4.3\%$ | $19.1 \pm 1.8\%$ | **$8.8 \pm 2.6\%$** |

**Table 3:** Model performance and approximation error for all methods and datasets when labels are corrupted using the *Periodic* noise function (average 30% label noise across all time steps). We report the clean test error (%) and mean approximation error of $\hat{Q}(t) \pm$ st.dev over 10 runs. The best-performing methods are highlighted in green. Continuous outperforms all baselines.

find that the temporal methods are consistently more accurate than their non-temporal counterparts, highlighting the importance of modeling temporal noise. For example, Plug-In is a temporal extension of Anchor, and we see that Plug-In outperforms Anchor in most settings. For example, on moving, the temporal method reduces the test error by over 10% compared to the nearest-performing static counterpart. This shows that our methods are robust to label noise despite being trained on data that is 30% corrupted with noisy labels, with no prior knowledge of the noise. This can have important consequences – e.g., in healthcare, where the noise is unknown and accurate models support life-or-death decisions.

Among the temporal methods, Continuous achieves the best performance in comparison to Plug-In and Discontinuous. Continuous' superiority is even clearer when compared to the static methods. Comparing the results for Table 2 (which assumes a Mixed noise model) and Table 3 (which assumes a Periodic noise model), we can see that these results hold across multiple types of temporal label noise. Fig. 2 demonstrates the ability of our methods to accurately reconstruct the underlying label noise. More importantly, the benefit of Continuous becomes more evident as the amount of noise increases in the data. In all these cases, we observe that temporal methods are consistently more robust to both temporal and static label noise. Overall, these findings suggest that we can improve performance by explicitly modeling how noise varies across time instead of assuming it is distributed uniformly in time.

**On Learning the Temporal Noise Function** Our results show that the performance increases hand in hand with our ability to estimate the noise model. In particular, we observe that low values of reconstruction consistently lead to low values of error rates.

We note that Continuous has the best reconstruction error (Approx. Error) on all datasets (Table 2). We qualitatively compare estimated noise functions and the ground truth for Continuous, Discontinuous,

and other baselines in Fig. 2. We use a DNN to model $Q$, which consistently estimates the noise function with lower mean absolute error across different families of noise functions. Appendix D.4 has extended results on all other datasets (including multi-class).

**On the Risk of Misspecification**    As shown in Section 3.2, our ability to learn under temporal label noise depends on our ability to correctly specify the temporal noise process. However, existing $Q$-estimators assume static, time-invariant label noise – which means that they will always fail to capture the correct noise function when it is temporal. In Fig. 5 (Appendix D), we validate this experimentally by showing how a model that assumes static noise compares to the *forward temporal loss* model learned using the true temporal noise function (see Appendix D.4 for other results). Our results highlight that there is no downside to modeling temporal effects. As shown, performance does not change when the temporal noise process is truly static. In contrast, performance drops when temporal effects exist. Using our temporal approaches they can relax this assumption to admit different types of temporal label noise. There is no cost to doing so, as our methods are competitive even if the noise model is truly static.

## 6    DEMONSTRATION ON STRESS DETECTION

We now demonstrate the viability of our approach on a real-world time series application where we have both clean *and* noisy labels. We focus on the task of stress detection, which is a common feature in modern smartwatches [9, 19] and is increasingly used to support interventions to mitigate clinical burnout [23, 29, 15]. Although many models use self-reported stress labels, these labels are subject to temporal label noise which arises from subjectivity, forgetfulness, and seasonal patterns in self-reported outcomes [53]. Given these effects, objective physiological measures (e.g., heart rate variability, temperature, etc.) are the standard for measuring stress and guiding interventions.

**Setup**    Our goal is to learn a time series classifier using the *noisy* labels (i.e., self-reported stress labels) that will generalize the *clean* labels (i.e., a physiological indicator of stress). We use a dataset from Goodday [24] monitoring stress in healthcare workers using physiological measures and self-reported measures. The dataset contains $n = 289$ unique individuals over $T = 50$ days. Each sequence has $d = 9$ features. Here, the *noisy label* is: $\tilde{y}_{i,t} = 1$ if person $i$ self-reports stress on day $t$. The *clean label* is $y_{i,t} = 1$ if the objective physiological measure of stress for person $i$ indicates stress on day $t$. We train sequential classifiers to predict stress over time. Our setup is identical to the one in Section 5, except that our training labels reflect real-world noisy labels and we have clean labels representing the ground truth stress.

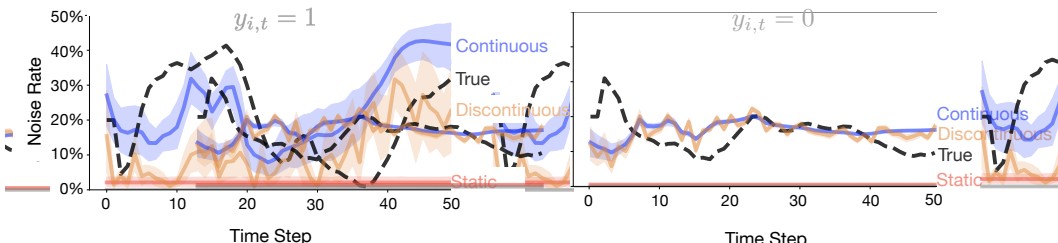

**Figure 3:** Temporal effects in label noise in a real-world stress detection task. We show noise rates when individuals are stressed (left) and not stressed (right). True noise rate is the average disagreement rates between clean and noisy labels over time. We can see clear, temporal label noise patterns – our temporal label noise methods do a superior job of approximating it.

**Results**    We report our results in Fig. 3 and Table 4. In Fig. 3, the black line indicates the average disagreement rates between the noisy self-reported label of stress and the clean physiological label over time. We can see clear temporal patterns in label noise, where participants under-report stress in a seasonal manner.

Our results are consistent with those in Section 5.2. In particular, we find that accounting for temporal label noise improves both training and test accuracy (Table 4). These results also highlight

| Method | What It Represents | Noise Estimation | Model Performance | |
|--------|--------------------|------------------|-------------------|---|
| | | Approx. Error | Train Error | Test Error |
| Ignore | ignoring label noise completely | $17.6 \pm 0.0\%$ | $31.0 \pm 0.3\%$ | $31.1 \pm 4.3\%$ |
| Static | static label noise correction per time step | $16.4 \pm 1.1\%$ | $31.0 \pm 0.3\%$ | $29.5 \pm 4.0\%$ |
| Discontinuous | temporal label noise correction with discontinuity | $\mathbf{11.4 \pm 0.8\%}$ | $\mathbf{27.1 \pm 0.7\%}$ | $\mathbf{26.0 \pm 2.5\%}$ |
| Continuous | temporal label noise correction imposing continuity | $\mathbf{9.7 \pm 1.3\%}$ | $\mathbf{21.1 \pm 0.5\%}$ | $\mathbf{25.5 \pm 1.7\%}$ |

**Table 4:** Noise rate estimation error and model performance for stress detection. We train sequential classifiers to predict stress using the training setup in Section 5. We report results for four methods to highlight the mechanism driving performance improvements through an ablation study. We show the approximation error noise rates (left); and the clear label error rate on the training sample (middle) and test sample (right) All values correspond to the mean 10-CV estimates $\pm$ st.dev.

the *mechanism* for these effects. Specifically, we see that lower approximation error leads to lower training error (via noise correction) and lower training error leads to lower test error (via generalization). In this case, all model classes are the same and the gains in Continuous are driven by its ability to learn the underlying temporal noise function in a joint optimization procedure.

**Discussion**  This demonstration shows the effectiveness of our methods in regimes without synthetic noise injection. We identified clean and noisy labels in this dataset using physiological measures and self-reports, respectively. In this scenario, temporal label noise represents the time-dependent process that models the discrepancy between these labels.

## 7 CONCLUDING REMARKS

Temporal label noise is an under-explored problem facing many real-world time series tasks. In such tasks, we must predict a sequence of labels, each of which can be subject to label noise whose distribution can change over time. Our work shows that existing methods for label noise substantially underperform when the distribution of label noise changes over time. As we show, we can achieve improved performance by accounting for temporal label noise. Given that this noise function is unknown, we also highlight the viability of learning the noise model from data. Finally, we perform a demonstration on a real-world source of temporal label noise – self-reported labels in digital health studies. In this setting, our methods continue to outperform methods that ignore such noise. Overall, temporal label noise presents a challenge in modeling time series – our paper underscores the necessity of explicitly modeling and accounting for temporal noise to achieve reliable and robust time series classification.

**Limitations**  We assume that all time series in a dataset have the same underlying label noise function. This may not be true as different annotators can introduce differing noise patterns. We can potentially relax this assumption by considering models that represent differences across annotators [see e.g., 63, 36, 70]. Additionally, we work with stationary time series, where the predictive distribution is stable over time. Though this is a standard assumption in time series work [see e.g., 80, 81, 78, 84], it is a limitation that can be addressed. For example, our loss-correction technique can be extended with learning theory from the study of non-stationary time series [33, 34]. Finally, the lack of time series datasets with *both* noisy and clean labels are few and far between. Without clean labels, it is difficult to verify that the temporal noise function learned is accurate or verify that a model trained on noisy labels generalizes to clean labels. We hope this paper will inspire the curation of more time series datasets containing both sets of labels.

**Acknowledgements** This work was supported by funding from the National Science Foundation IIS 2040880, IIS 2313105, and the NIH Bridge2AI Center Grant U54HG012510. SN was supported by a CIHR Vanier Scholarship. AG is a CIFAR AI Chair. Resources used in preparing this research were provided, in part, by the Province of Ontario, the Government of Canada through CIFAR, and companies sponsoring the Vector Institute. We also thank Abhishek Moturu and Vijay Giri for valuable discussions about the project.

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

# Supplementary Materials

## A PROOFS

In what follows, we use vector notation for completeness and clarity of exposition. We describe key quantities and notation that we use in Table 5. We will also use Assumption 1 and Assumption 2 to factor the noisy label distribution as follows:

$$p(\tilde{\mathbf{y}}_{1:T} \mid \mathbf{x}_{1:T}) = \prod_{t=1}^{\top} q_t(\tilde{\mathbf{y}}_t \mid \mathbf{x}_{1:t}). \tag{6}$$

| Notation | Description |
|---|---|
| $\boldsymbol{p}(\mathbf{y}_t \mid \mathbf{x}_{1:t}) := \left[ p(\mathbf{y}_t = c \mid \mathbf{x}_{1:t}) \right]_{c=1:C}^{\top}$ | Vector of probabilities for each label value, for the clean label distribution $\boldsymbol{p}(\mathbf{y}_t \mid \mathbf{x}_{1:t}) \in \mathbb{R}^{C \times 1}$ |
| $\boldsymbol{p}(\tilde{\mathbf{y}}_t \mid \mathbf{x}_{1:t}) := \left[ p(\mathbf{y}_t = c \mid \mathbf{x}_{1:t}) \right]_{c=1:C}^{\top}$ | Vector of probabilities for each possible label value, for the noisy label distribution $\boldsymbol{p}(\tilde{\mathbf{y}}_t \mid \mathbf{x}_{1:t}) \in \mathbb{R}^{C \times 1}$ |
| $\boldsymbol{h}_\theta(\boldsymbol{x}_{1:t}) = \boldsymbol{p}_\theta(\mathbf{y}_t \mid \mathbf{x}_{1:t} = \boldsymbol{x}_{1:t})$ | Classifier that predicts label distribution at $t$ given preceding observations $\mathbb{R}^{d \times t} \to \mathbb{R}^C$ |
| $\boldsymbol{h}_\theta(\boldsymbol{x}_{1:t}) = \boldsymbol{\psi}^{-1}(\boldsymbol{g}_\theta(\boldsymbol{x}_{1:t}))$ | When $h_\theta$ is a deep network, $g_\theta$ is the final logits, and $\boldsymbol{\psi} : \Delta^{C-1} \to \mathbb{R}^C$ represents an invertible link function (e.g., softmax) |
| $\boldsymbol{Q}_t := \left[ q_t(\tilde{\mathbf{y}}_t = k \mid \mathbf{y}_t = j) \right]_{j,k}$ | The temporal noise matrix at time $t$: $\boldsymbol{Q}_t \in \mathbb{R}^{C \times C}$ |
| $\ell_t(y_t, \boldsymbol{h}_\theta(\boldsymbol{x}_{1:t})) = -\log p_\theta(\mathbf{y}_t = y_t \mid \mathbf{x}_{1:t} = \boldsymbol{x}_{1:t})$ | Loss at $t$: $\mathcal{Y} \times \mathbb{R}^C \to \mathbb{R}$ |
| $\ell_{\psi,t}(y_t, \boldsymbol{h}_\theta(\boldsymbol{x}_{1:t})) = \ell_t(y_t, \boldsymbol{\psi}^{-1} \boldsymbol{h}_\theta(\boldsymbol{x}_{1:t}))$ | A composite loss function using a link function $\psi$ |
| $\boldsymbol{\ell}_t(\boldsymbol{h}_\theta(\boldsymbol{x}_{1:t})) = \left[ \ell_t(c, \boldsymbol{h}_\theta(\boldsymbol{x}_{1:t})) \right]_{c=1:C}^{\top}$ | Vector of NLL losses for each possible value of the ground truth $\mathbb{R}^C \to \mathbb{R}^C$ |
| $\overrightarrow{\ell}_{t,\psi}(c, \boldsymbol{h}_\theta(\boldsymbol{x}_{1:t})) = \ell_t(c, \boldsymbol{Q}_t^\top \cdot \boldsymbol{\psi}^{-1}(\boldsymbol{g}_\theta))$ | Forward loss for class $c$ |
| $\overrightarrow{\ell}_{seq,\psi}(\boldsymbol{y}_{1:T}, \boldsymbol{h}_\theta(\boldsymbol{x}_{1:t})) = \sum_{t=1}^{T} \overrightarrow{\ell}_{t,\psi}(c, \boldsymbol{h}_\theta(\boldsymbol{x}_{1:t}))$ | Sequence forward loss |

**Table 5:** Quantities and definitions used for the following proof

**Proof of Proposition 1**

*Proof.* Our goal is to show:

$$\operatorname*{argmin}_{\theta} \mathbb{E}_{\tilde{\mathbf{y}}_{1:T}, \mathbf{x}_{1:T}} \overrightarrow{\ell}_{seq,\psi}(\mathbf{y}_{1:T}, \boldsymbol{g}_\theta(\mathbf{x}_{1:T})) = \operatorname*{argmin}_{\theta} \sum_{t=1}^{\top} \mathbb{E}_{\mathbf{y}_{1:t}, \mathbf{x}_{1:t}} \ell_{t,\phi}(\mathbf{y}_{1:T}, \boldsymbol{g}_\theta(\mathbf{x}_{1:T})).$$

First, note that:

$$\overrightarrow{\ell}_{t,\psi}(\mathbf{y}_t, \boldsymbol{h}_\theta(\mathbf{x}_{1:t})) = \ell_t(\mathbf{y}_t, \boldsymbol{Q}_t^\top \boldsymbol{\psi}^{-1}(\boldsymbol{g}_\theta(\mathbf{x}_{1:t}))) \tag{7}$$
$$= \ell_{\boldsymbol{\phi}_t,t}(\mathbf{y}_t, \boldsymbol{g}_\theta(\mathbf{x}_{1:t})), \tag{8}$$

where $\boldsymbol{\phi}_t^{-1} = \boldsymbol{\psi}^{-1} \circ \boldsymbol{Q}_t^\top$. Thus, $\boldsymbol{\phi}_t : \Delta^{C-1} \to \mathbb{R}^C$ is invertible, and is thus a proper composite loss [58].

Thus, as shown in Patrini et al. [50]:

$$\operatorname*{argmin}_{\theta} \mathbb{E}_{\tilde{y}_t, \mathbf{x}_{1:t}} \ell_{\phi,t}(y_t, \boldsymbol{g}_\theta(\mathbf{x}_{1:t})) = \operatorname*{argmin}_{\theta} \mathbb{E}_{\tilde{y}_t | \mathbf{x}_{1:t}} \ell_{\phi_t, t}(y_t, \boldsymbol{g}_\theta(\mathbf{x}_{1:t})) \tag{9}$$

$$= \phi_t(\boldsymbol{p}(\tilde{y}_t \mid \mathbf{x}_{1:t})) \qquad \text{(property of proper composite losses)}$$

$$= \boldsymbol{\psi}((\boldsymbol{Q}_t^{-1})^\top \boldsymbol{p}(\tilde{y}_t \mid \mathbf{x}_{1:t}))) \tag{10}$$

$$= \boldsymbol{\psi}(\boldsymbol{p}(y_t \mid \mathbf{x}_{1:t})) \tag{11}$$

The above holds for the minimizer at a single time step, not the sequence as a whole. To find the minimizer of the loss over the entire sequence:

$$\operatorname*{argmin}_{\theta} \mathbb{E}_{\mathbf{x}_{1:T}, \tilde{\mathbf{y}}_{1:T}} \overrightarrow{\ell}_{seq, \psi}(\tilde{\mathbf{y}}_{1:T}, \boldsymbol{g}_\theta(\mathbf{x}_{1:T})) = \operatorname*{argmin}_{\theta} \mathbb{E}_{\tilde{\mathbf{y}}_{1:T} | \mathbf{x}_{1:T}} \overrightarrow{\ell}_{seq, \psi}(\tilde{\mathbf{y}}_{1:T}, \boldsymbol{g}_\theta(\mathbf{x}_{1:T})) \tag{12}$$

$$= \operatorname*{argmin}_{\theta} \mathbb{E}_{\tilde{\mathbf{y}}_{1:T} | \mathbf{x}_{1:T}} \sum_{t=1}^{\top} \overrightarrow{\ell}_{t, \psi}(\tilde{y}_t, \boldsymbol{g}_\theta(\mathbf{x}_{1:t})) \tag{13}$$

$$= \operatorname*{argmin}_{\theta} \sum_{t=1}^{\top} \mathbb{E}_{\tilde{\mathbf{y}}_{1:T} | \mathbf{x}_{1:T}} \overrightarrow{\ell}_{t, \psi}(\tilde{y}_t, \boldsymbol{g}_\theta(\mathbf{x}_{1:t})) \tag{14}$$

$$= \operatorname*{argmin}_{\theta} \sum_{t=1}^{\top} \mathbb{E}_{\tilde{y}_t | \mathbf{x}_{1:t}} \overrightarrow{\ell}_{t, \psi}(\tilde{y}_t, \boldsymbol{g}_\theta(\mathbf{x}_{1:t})) \tag{15}$$

$$= \operatorname*{argmin}_{\theta} \sum_{t=1}^{\top} \mathbb{E}_{\tilde{y}_t | \mathbf{x}_{1:t}} \ell_{t, \phi}(\tilde{y}_t, \boldsymbol{g}_\theta(\mathbf{x}_{1:t})) \tag{16}$$

As the minimizer of the sum will be the function that minimizes each element of the sum, then $\operatorname{argmin}_\theta \mathbb{E}_{\tilde{\mathbf{y}}_{1:T}, \mathbf{x}_{1:T}} \overrightarrow{\ell}_{seq, \psi}(\mathbf{y}_{1:T}, \boldsymbol{g}_\theta(\mathbf{x}_{1:T})) = \boldsymbol{\psi}(\boldsymbol{p}(\mathbf{y}_{1:T} \mid \mathbf{x}_{1:T}))$. Note that the $\operatorname{argmin}_\theta \sum_{t=1}^{\top} \mathbb{E}_{\mathbf{y}_{1:t}, \mathbf{x}_{1:t}} \ell_{t, \phi}(\mathbf{y}_{1:T}, \boldsymbol{g}_\theta(\mathbf{x}_{1:T})) = \boldsymbol{\psi}(\boldsymbol{p}(\mathbf{y}_{1:T} \mid \mathbf{x}_{1:T}))$, because the minimizer of the NLL is the data distribution. Thus, $\operatorname{argmin}_\theta \mathbb{E}_{\tilde{\mathbf{y}}_{1:T}, \mathbf{x}_{1:T}} \overrightarrow{\ell}_{seq, \psi}(\mathbf{y}_{1:T}, \boldsymbol{g}_\theta(\mathbf{x}_{1:T})) = \operatorname{argmin}_\theta \sum_{t=1}^{\top} \mathbb{E}_{\mathbf{y}_{1:t}, \mathbf{x}_{1:t}} \ell_{t, \phi}(\mathbf{y}_{1:T}, \boldsymbol{g}_\theta(\mathbf{x}_{1:T}))$.

$\square$

## B  CONTINUOUS LEARNING ALGORITHM

We summarize the augmented lagrangian approach to solving the continuous objective in Algorithm 1

---

**Algorithm 1** Continuous Learning Algorithm

---

**Input:** Noisy Training Dataset $D$, hyperparameters $\gamma$ and $\eta$
**Output:** Model $\theta$, Temporal Noise Function $\omega$
    $c \leftarrow 1$ and $\lambda \leftarrow 1$
    **for** $k = 1, 2, 3, \ldots,$ **do**
        $\theta^k, \omega^k = \arg\min_{\theta, \omega} \mathcal{L}(\theta, \omega)$         $\triangleright$ Computed with SGD using the Adam optimizer
        $\lambda \leftarrow \lambda + c * R_t(\theta^k, \omega^k)$         $\triangleright$ Update Lagrange multiplier
        **if** $k > 0$ and $R_t(\theta^k, \omega^k) > \gamma R_t(\theta^{k-1}, \omega^{k-1})$ **then**
            $c \leftarrow \eta c$
        **else**
            $c \leftarrow c$
        **end if**
        **if** $R_t(\theta^k, \omega^k) == 0$ **then**
            break
        **end if**
    **end for**

---

For all experiments we set $\lambda = 1, c = 1$, $\gamma = 2$, and $\eta = 2$. $k$ and the maximum number of SGD iterations are set to 15 and 10, respectively. This is to ensure that the total number of epochs is 150, which is the max number of epochs used for all experiments.

## C  PLUG-IN PROCEDURE

1. Fit a probabilistic classifier to predict noisy labels from the observed data.
2. For each class $y \in \mathcal{Y}$ and time $t \in [1 \dots T]$:
    i  Identify anchor points for class $y$: $\bar{\boldsymbol{x}}_t^j = \arg\max_{\boldsymbol{x}_t} p(\tilde{\mathrm{y}}_t = y \mid \boldsymbol{x}_{1:t})$.
    ii  Set $\hat{\boldsymbol{Q}}(t)_{y,y'}$ as the probability of classifier predicting class $y'$ at time $t$ given $\bar{\boldsymbol{x}}_t^j$.

## D  SUPPORTING MATERIAL FOR EXPERIMENTAL RESULTS

In what follows, we describe details for reproducing the experiments shown in the paper. Our code is available on GitHub.

### D.1  DATASET DETAILS

| Dataset | Classification Task | $n$ | $d$ | $T$ |
|---|---|---|---|---|
| `blinking` [60] | Eye Open vs Eye Closed | 299 | 14 | 50 |
| `sleeping` [22] | Sleep vs Awake | 964 | 7 | 100 |
| `moving` [59] | Walking vs Not Walking | 192 | 9 | 50 |
| `senior` [44] | Walking vs Not Walking | 444 | 6 | 100 |

**Table 6:** Datasets used in the experiments. Classification tasks, number of samples ($n$), dimensionality at each time step ($d$), and sequence length ($T$) are shown.

**moving**    from UC Irvine [59] consists of inertial sensor readings of 30 adult subjects performing activities of daily living. The sensor signals are already preprocessed and a vector of features at each time step are provided. We apply z-score normalization at the participant-level, then split the dataset into subsequences of a fixed size 50. We use a batchsize of 64.

**senior**    from UC Irvine [44] consists of inertial sensor readings of 18 elderly subjects performing activities of daily living. The sensor signals are already preprocessed and a vector of features at each time step are provided. We apply z-score normalization at the participant-level, then split the dataset into subsequences of a fixed size 100. We use a batchsize of 256.

**sleeping**    from Physionet [22] consists of EEG data measured from 197 different whole nights of sleep observation, including awake periods at the start, end, and intermittently. We apply z-score normalization at the whole night-level. Then downsample the data to have features and labels each minute, as EEG data is sampled at 100Hz and labels are sampled at 1Hz. We then split the data into subsequences of a fixed size 100. We use a batchsize of 512.

**blinking**    from UC Irvine [60] consists of data measured from one continuous participant tasked with opening and closing their eyes while wearing a headset to measure their EEG data . We apply z-score normalization for the entire sequence, remove outliers (>5 SD away from mean), and split into subsequences of a fixed size 50. We use a batchsize of 128.

**synthetic**    We generate data for binary and multiclass classification with $n = 1\,000$ samples and $d = 50$ features over $T = 100$ time steps. We generate the class labels and obvservations for each time step using a Hidden Markov Model (HMM). The transition matrix generating the markov chain is uniform ensuring an equal likelihood of any state at any given time. We corrupted them using multidimensional (50) Gaussian emissions. The mean of the gaussian for state/class $c$ is set to $c$

with variance 1.5 (i.e. class 1 has mean 1 and variance 1.5). The high-dimensionality and overlap in feature-space between classes makes this a sufficiently difficult task, especially under label noise. We use a batchsize of 256

## D.2 NOISE INJECTION AND NOISE REGIMES

We create noise labels for each dataset by applying one of the six temporal noise functions shown in Fig. 4. We choose functions to capture different functional forms, assumptions on time-variance, and class-dependence.

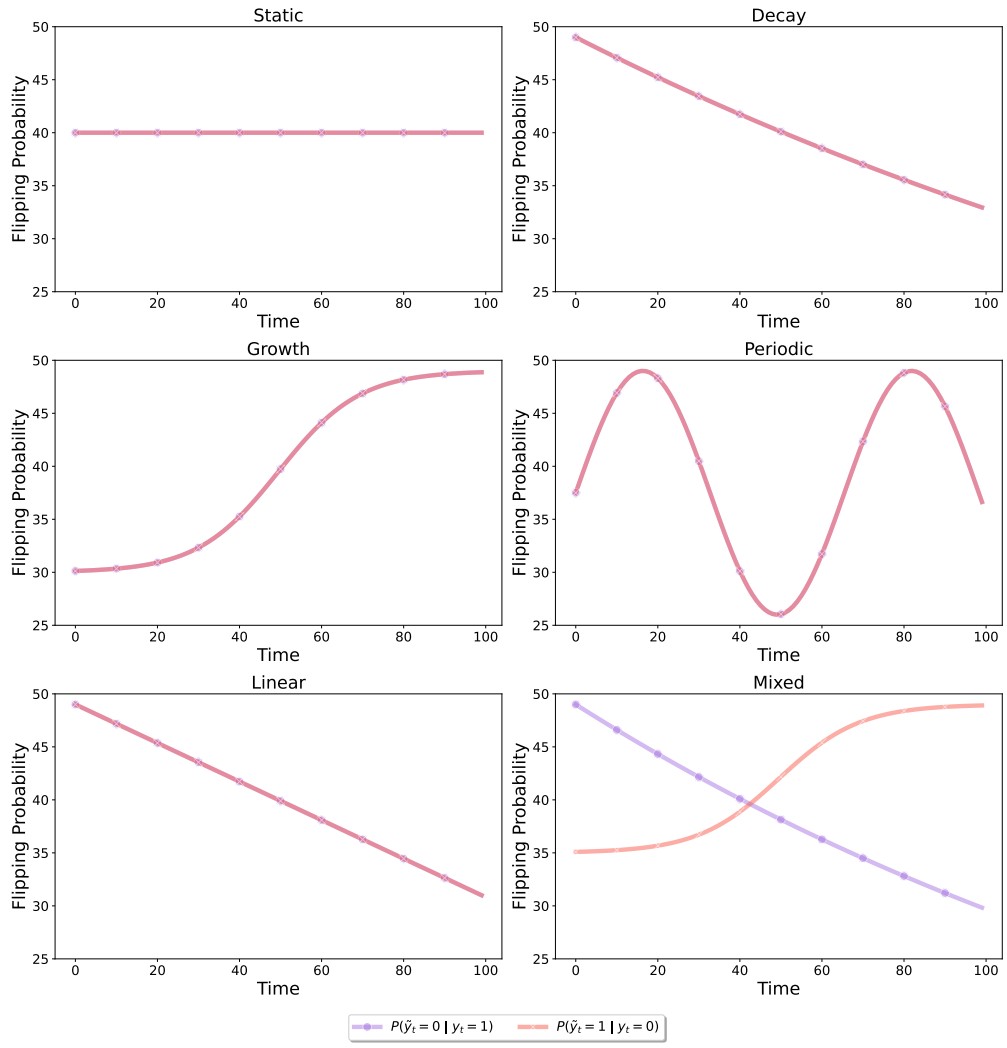

**Figure 4:** Temporal label noise function $Q(t)$ used in the experiments. We present six examples for binary classification task (from top-left clockwise): static, decay periodic, mixed, linear, growth. Each plot shows the off-diagonal entries of various parameterized forms of $Q(t)$.

### D.3 DETAILS ON TRAINING ALGORITHMS

**GRU**    the GRU $r : \mathbb{R}^d \times \mathbb{Z} \to \mathbb{R}^C \times \mathbb{Z}$ produces an *output vector* such that the output of $r(\boldsymbol{x}_t, \boldsymbol{z}_{t-1})$ is our model for $\boldsymbol{h}_\theta(\boldsymbol{x}_{1:t})$, and a *hidden state* $\boldsymbol{z}_t \in \mathbb{Z}$ that summarizes $\boldsymbol{x}_{1:t}$. We use a softmax activation on the output vector of the GRU to make it a valid parameterization of $\boldsymbol{p}_\theta(y_t \mid \boldsymbol{x}_{1:t})$. The GRU has a single hidden layer with a 32 dimension hidden state.

**Continuous**   Continuous uses an additional fully-connected neural network with 10 hidden layers that outputs a $C * C$-dimensional vector to represent each entry of a flattened $\hat{\boldsymbol{Q}}_t$. To ensure the output of this network is valid for Def. 1, we reshape the prediction to be $C \times C$, apply a row-wise softmax function, add this to the identity matrix to ensure diagonal dominance, then rescale the rows to be row-stochastic. These operations are all differentiable, ensuring we can optimize this network with standard backpropagation.

**VolMinNet and Discontinuous**   We do a similar parameterization for VolMinNet and Discontinuous, using a set of differentiable weights to represent the entries of $\boldsymbol{Q}_t$ rather than a neural network.

**Anchor and Plug-In**   Patrini et al. [50] show that in practice taking the 97th percentile anchor points rather than the maximum yield better results, so we use that same approach in our experiments. They also describe a two-stage approach: 1) estimate the anchor points after a warmup period 2) use the anchor points to train the classifier with forward corrected loss. We set the warmup period to 25 epochs.

**Hyperparameters**   We train each model for 150 epochs using the adam optimizer with default parameters and a learning rate of 0.01. We train all models using the same set of hyperparameters for experiment, set the batch size manually for each dataset, and avoid hyperparameter tuning to avoid label leakages. For VolMinNet, Discontinuous, and Continuous we use adam optimizer with default parameters and a learning rate of 0.01 to optimize each respective $\hat{\boldsymbol{Q}}_t$-estimation technique. $\lambda$ was set to $1e-4$ for VolMinNet and Discontinuous for all experiments following the setup of Li et al. [38].

### D.4 ADDITIONAL RESULTS

#### D.4.1 STATIC APPROXIMATION

Here we study the performance of *forward temporal loss* where we know the noise function $Q(t)$ – that is, even if we could *perfectly* estimate the noise process – and where we have a static estimate of $Q(t)$ (the average over time). We find that even if the noise process is perfectly estimated, accounting for temporal noise outperforms a static estimate.

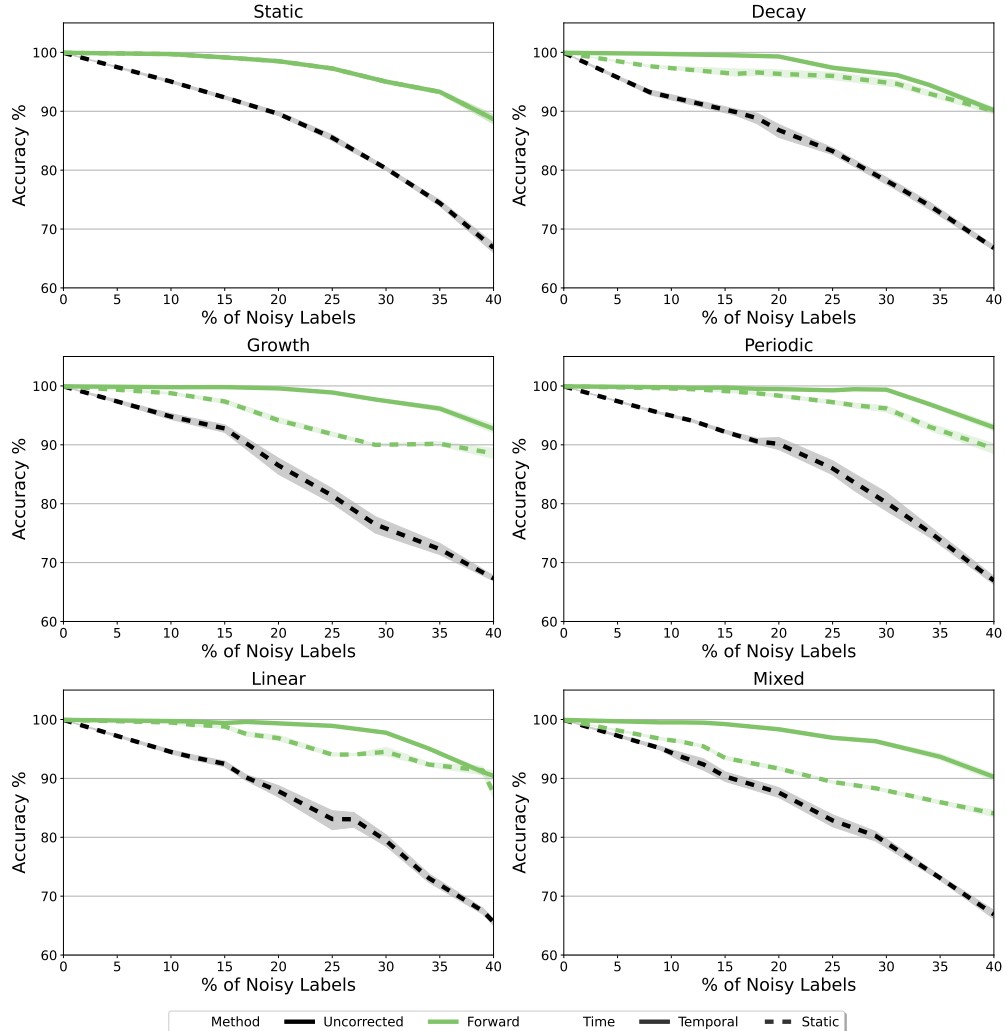

**Figure 5:** Comparing performance of models trained with *forward temporal loss* vs no noise correction on synth with varying degrees of temporal label noise using either the true temporal noise function (Temporal) or the average temporal noise function (Static). Error bars are st. dev. over 10 runs.

### D.4.2 MULTICLASS CLASSIFICATION

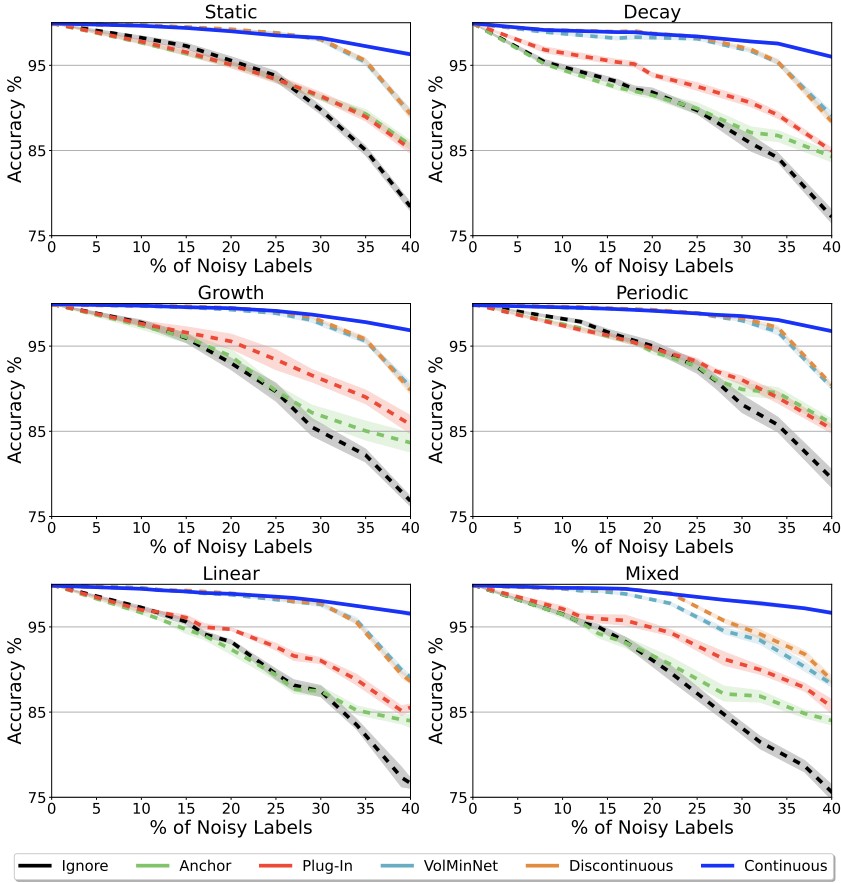

**Figure 6:** Comparison of clean test set Accuracy (%) for `synth` across varying degrees of temporal label noise comparing all methods for 3-class classification. Error bars are st. dev. over 10 runs.

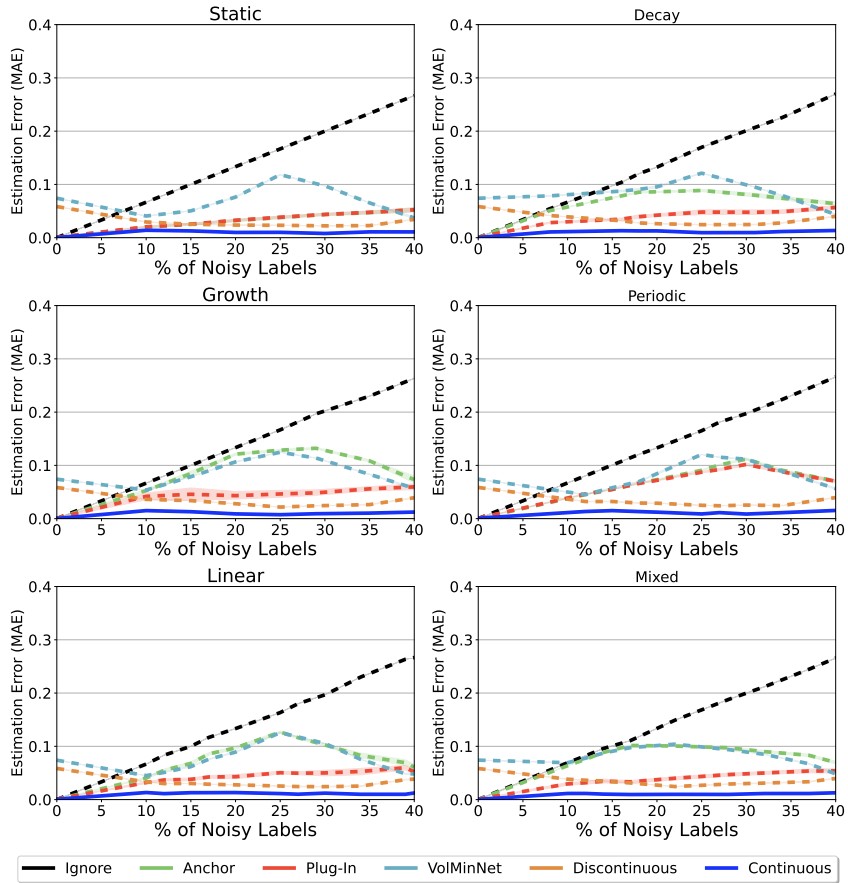

**Figure 7:** Comparison of noisy function reconstruction Mean Absolute Error (MAE) for `synth` across varying degrees of temporal label noise comparing all methods for 3-class classification. Error bars are st. dev. over 10 runs.

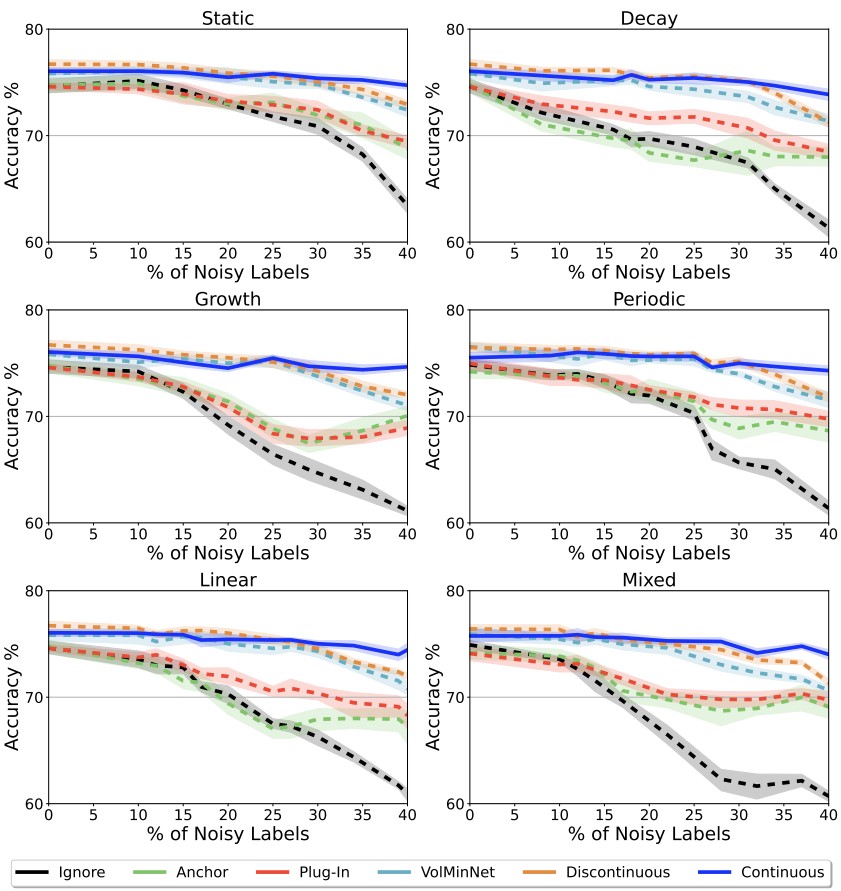

**Figure 8:** Comparison of clean test set Accuracy (%) for `sleeping` across varying degrees of temporal label noise comparing all methods for 3-class classification. Error bars are st. dev. over 10 runs.

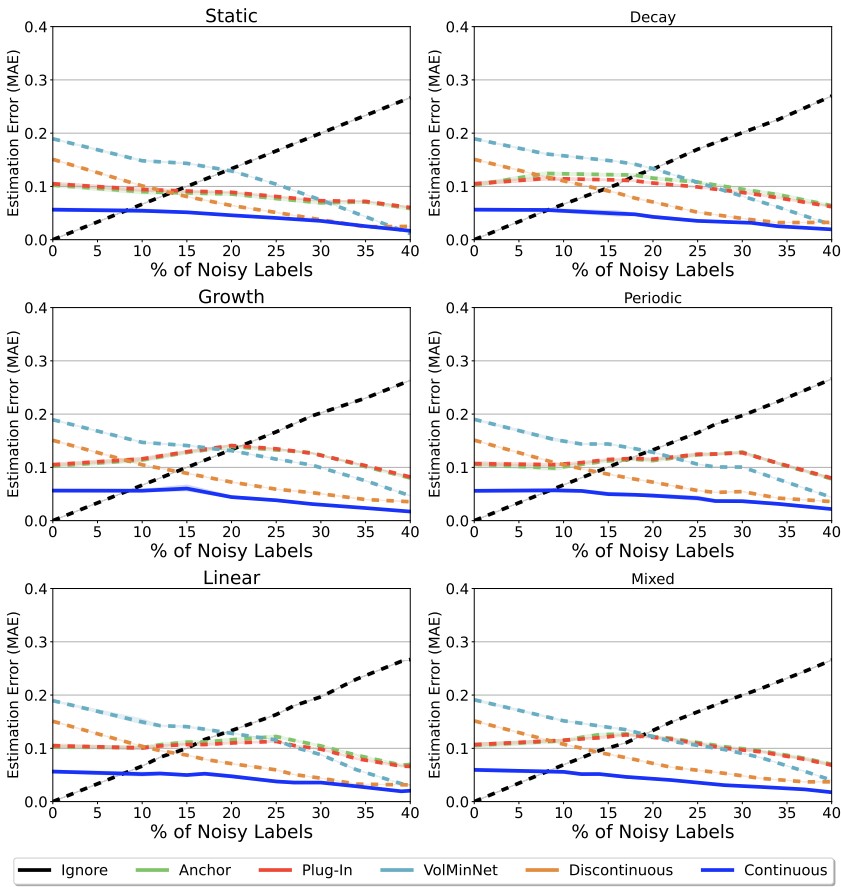

**Figure 9:** Comparison of noisy function reconstruction Mean Absolute Error (MAE) for `sleeping` across varying degrees of temporal label noise comparing all methods for 3-class classification. Error bars are st. dev. over 10 runs.

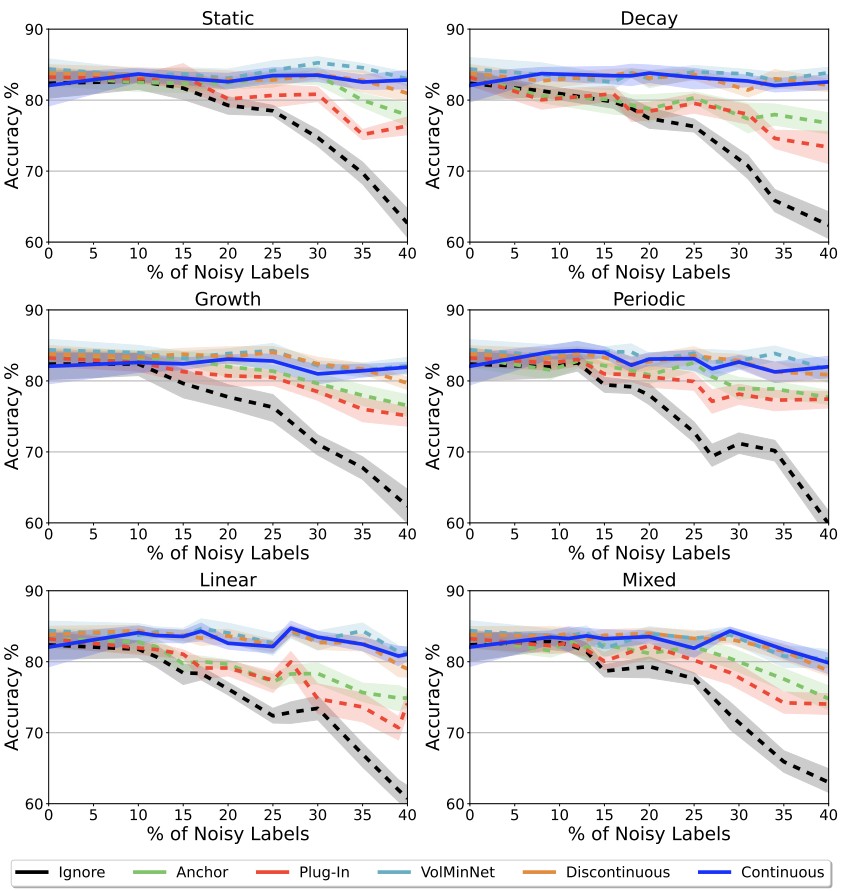

**Figure 10:** Comparison of clean test set Accuracy (%) for `moving` across varying degrees of temporal label noise comparing all methods for 4-class classification. Error bars are st. dev. over 10 runs.

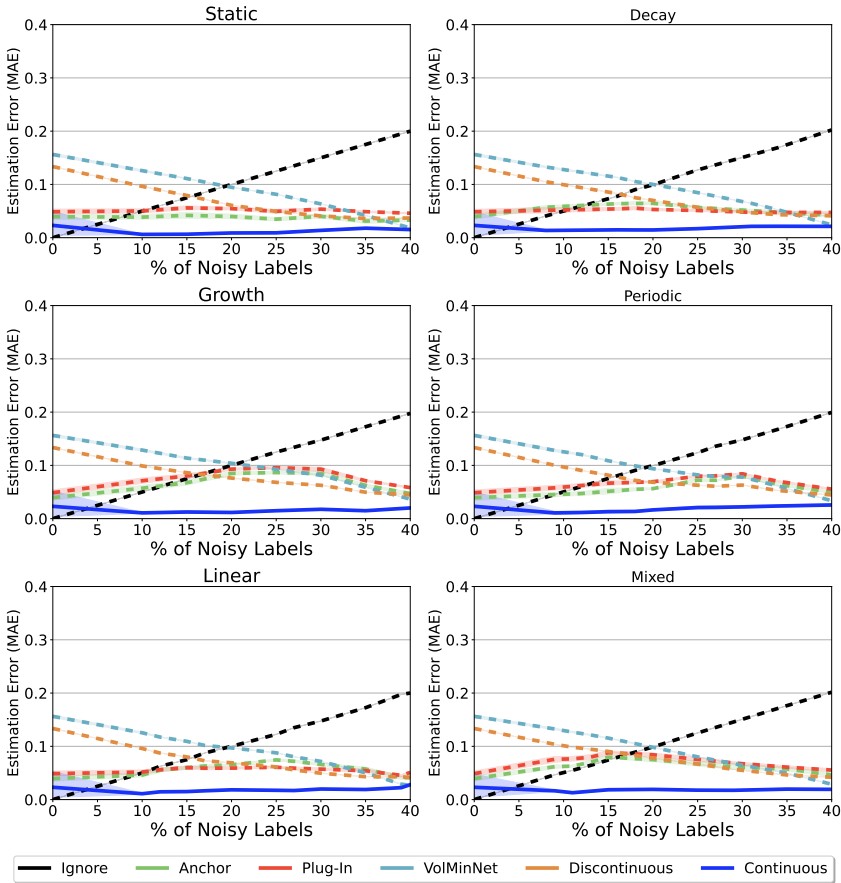

**Figure 11:** Comparison of noisy function reconstruction Mean Absolute Error (MAE) for `har` across varying degrees of temporal label noise comparing all methods for 4-class classification. Error bars are st. dev. over 10 runs.

