# OpenReview forum: "Learning under Temporal Label Noise"
_ICLR.cc/2025/Conference — ICLR 2025 Poster_

### Official Review · Reviewer_EoQ6 · 2024-10-27

**Soundness:** 3
**Presentation:** 3
**Contribution:** 2
**Rating:** 5
**Confidence:** 3

**Summary:**

The paper introduces the problem of temporal label noise, where label quality fluctuates over time due to time-dependent noise. This type of noise may improve, deteriorate, or change periodically, impacting the accuracy of predictions. The authors define this problem and highlight that existing methods fail to handle the temporal aspect of label noise effectively. They propose methods to estimate the time-varying noise function directly from data, allowing the development of noise-tolerant classifiers. Experiments on real-world datasets validate their proposal, demonstrating its effectiveness under various types of temporal label noise.

**Strengths:**

1. The problem of Temporal Label Noise is both interesting and important, and this paper is the first to formally define it.

2. The paper is well-organized, clearly written, and easy to follow.

3. The mathematical symbols and equations are well-defined and easy to understand.

**Weaknesses:**

1. My primary concern is with the problem formulation, or more specifically, the goal of the problem. In Section 2.1, the paper aims to "estimate parameters $\hat{\theta}$ for a model robust to noise," which implicitly assumes the existence of a **fixed and static optimal model** $\theta^\star$—i.e., that p(y_t|x_t) remains consistent across all timestamps. However, in time-series problems, the environment is constantly changing, and the optimal model for different timestamps should also be changing.

2. The other major concern relates to the methodology. The paper appears to assume that the function Q is known to the learner, and focuses only on optimizing the parameter w. However, I think that **obtaining an appropriate function Q is the most challenging aspect of this problem**. Although in Section 3.2, the authors select Q as a fully connected neural network as a general form for the practical scenarios, it seems to be inconsistent with the earlier discussions in Table 1.

3. Another concern is about the experiments. The authors seem to only conduct experiments on low-dimensional datasets. I think that including experiments on larger-sized datasets would strengthen the results and make the findings more convincing.

Please let me know if I have misunderstood any part of the paper.

**Questions:**

See Weaknesses above

---

> ### Author Response · Authors · 2024-11-22
> **Response to Reviewer EoQ6**
>
> Thanks for your time and detailed feedback! We've addressed most of the questions and comments below. If there is anything else, please let us know!
>
>
> > **W1) My primary concern is with the problem formulation, or more specifically, the goal of the problem…in time-series problems, the environment is constantly changing, and the optimal model for different timestamps should also be changing.**
>
> This appears to be a misunderstanding. Time series indeed represent changing environments, but it’s overwhelmingly common to assume the *relationship* between inputs and outputs is fixed because the outputs can still depend on the inputs (e.g., ARIMA makes this assumption). Any ML model trained to predict a label based on a window of time series makes this assumption.
>
> If this question references non-stationary time series, where the generative model changes over time, then this is out of our scope. Many domains feature stationary time series and most methods assume stationarity, which allows for clearer analysis and is often sufficient for practical use. Most state-of-the-art approaches in time-series assume stationarity, which allows for clearer theoretical analysis, and does not present a problem in practice (see e.g.,  recent papers from top conferences [NeurIPS 2024](https://arxiv.org/abs/2411.01842), [NeurIPS 2024](https://openreview.net/forum?id=UE6CeRMnq3&noteId=7LmUCl1XoB), [ICML 2024](https://proceedings.mlr.press/v235/woo24a.html), [ICLR 2024](https://openreview.net/forum?id=bWcnvZ3qMb))
>
> For further empirical evidence, we see from our own experiments on several real-world time-series datasets that we learn near-optimal models without considering non-stationarity.
>
> > **W2) The paper appears to assume that the function Q is known to the learner, and focuses only on optimizing the parameter w.**
>
> We want to flag a misunderstanding here: **we never assume  $Q$  is known to the learner**. The only assumption the learner needs is about the class of functions parameterizing $Q$ (e.g., DNNs).  $Q$  is parameterized by  $\omega$ , which is learned *entirely* from the noisy labels, without any prior assumptions about its specific functional form. The true underlying $Q$ is known only to us, the authors, as it allows us to measure how well our proposed methods can learn $Q$ - we report these results in Table 2 and 3 (see Approx. Error).
>
> The entire purpose of Section 3 is to construct methods that can learn $Q$ from data. To ensure this point is clear, we have modified the text to better motivate this section and outline the assumptions we make (and, more importantly, do not make).
>
> We hope this clarifies any confusion! Please let us know if this is not the case.
>
> > **W3) The authors seem to only conduct experiments on low-dimensional datasets. I think that including experiments on larger-sized datasets would strengthen the results and make the findings more convincing.**
>
> We disagree - these datasets are of a reasonable size as each instance is roughly 100 time steps long. So for each dataset, we have NxT instances with each instance having D dimensions (e.g., Sleep dataset contains 96,400 unique instances, each with 7 features and a label). Additionally, these datasets have been widely adopted as benchmarks in time-series research and are not exclusive to this paper.
>
> That said, if there are bigger time-series datasets (with sequences of labels at each time-step) you would have liked to see included, please let us know, and we will do our best to incorporate them in a revised version during the discussion period!
>
> —
>
> We hope to have clarified any misunderstandings and questions you had. However, if there are any other questions we can answer in the discussion period, please let us know and we are happy to provide more details.
>
> Thank you again!

---

> ### Comment · Reviewer_EoQ6 · 2024-11-24
>
> Thank you to the authors for their response.
>
> Regarding Q1, I recommend that the authors include the references in the revised version. Additionally, it would be beneficial to include a discussion on applying the method to handle non-stationary environments.
>
> For Q2, my concern remains unresolved. Specifically, the statement "The only assumption the learner needs is about the class of functions parameterizing Q" suggests that the learner must know the type of function. However,  I think that identifying an appropriate class of functions for Q is the most challenging aspect of this problem.

---

> > ### Author Response · Authors · 2024-11-24
> > **Response to Reviewer EoQ6**
> >
> > Thank you for your response!. Following your suggestion for W1, we have added the references and discussion about non-stationary environments in the Limitations and Related Work (Section 6) — please see the latest revision. With this change, we are glad to notice that W1 and W3 are resolved.
> >
> > Regarding W2, we believe there is still a misunderstanding:
> >
> > * **When we say**: "The only assumption the learner needs is about the class of functions parameterizing Q"
> > * **What we mean to say is**: “A practitioner has to decide on what type of *algorithm* they want to use to model Q.”
> >   *  This is not an *assumption* but rather a *design decision*.
> >   *  This decision is part of *any* ML pipeline, where practitioners decide on the type of model to train based on the problem domain.
> > For example, choosing a Transformer vs an LSTM for a language-modeling task.
> >   *  **This is a core property of *all* ML and is not unique to our paper.**
> >
> > For further evidence, in our paper, we use a DNN to model Q. This can capture *any* temporal noise function because DNNs are universal function approximators. The DNN *learns* Q directly from data, with no prior assumptions about the temporal noise function, and it works well for all the types of temporal label noise we use in the dataset - we have clarified this in Section 3.4.
> >
> > We hope that this clarifies this point — please let us know if not!
> >
> > Thank you again for your engagement during the Discussion period.

---

> > > ### Comment · Reviewer_EoQ6 · 2024-11-25
> > >
> > > Thank you to the authors for the response. It appears that I did not misunderstand the paper: the function type of Q is indeed known to the learner, and the learner is only required to learn the parameter w. While this deisgn poses no significant issues for previous works that used this design to learn a classifier (from a class of functions), the authors' statement, "As shown in Table 1, we can capture a wide variety of temporal noise using this representation" (line 186), seems somewhat overstated. Specifically, it implies the ability to capture **any** type of noise function, which may not be the case (the type of noise function is **known**, not **learned**). I encourage the authors to clarify this point in the revised version to avoid potential overclaims.
> > >
> > > Overall, the response has addressed most of my concerns, and I have decided to raise my score accordingly.

---

### Official Review · Reviewer_qqUX · 2024-11-03

**Soundness:** 4
**Presentation:** 4
**Contribution:** 3
**Rating:** 8
**Confidence:** 3

**Summary:**

The paper proposes a model to incorporate varying noise rates for time-series classification. The method specifies a matrix-valued function indicating label-noise distribution. This noise function is incorporated into a forward temporal loss over noisy labels that is used to learn a classifier (using a neural net) minimizing the loss. The authors describe three approaches: continuous, discontinuous, and plug-in. They empirically evaluate these approaches in various benchmark datasets. They also demonstrate the approach in a real-world dataset within the healthcare domain in which the authors claim that the problem of temporal noisy labels is prevalent in time-series classification. The empirical results clearly indicate that the proposed approach works well compared to SOTA static approaches that treat noisy labels as static rather than temporal.

**Strengths:**

1) The formulation of the proposed methodology is clear and concise.
2) The motivation and subsequent real-world example is well demonstrated.

**Weaknesses:**

(1) It is not clear how is learning achieved when weights are initialized for each time step in discontinuous estimation. It looks like the dataset is small for the learning models to converge as all the datasets largely seem to have less than 1000 samples.
(2) Can the estimator in the discontinuous case be replaced by a simpler model (with lesser number of parameters)? If so, can such an alternative be used as baseline mechanisms to compare for temporal noise (as all alternative mechanisms shown in the paper are for static cases)?
(3) It is interesting that both continuous and discontinuous estimations perform quite similarly on the real-world stress detection example. Does this mean that the proposed approach works best when there is large variance in noise rates?

**Questions:**

(1) Is there an assumption that all data instances have equal number of time steps?

---

> ### Author Response · Authors · 2024-11-22
> **Response to Reviewer qqUX**
>
> Thanks for your time and detailed feedback! We've addressed most of the questions and comments below. If there is anything else, please let us know!
>
> > **Q1) How is learning achieved when weights are initialized for each time step in discontinuous estimation / Can the estimator in the discontinuous case be replaced by a simpler model (with lesser number of parameters)?**
>
> The discontinuous case is actually the simplest possible model we can use. In this case, we are just learning the specific entries for the temporal noise function (a C \times C matrix) at each time-step $t$. For example, in the binary-setting each time-step will only have 2 parameters we are estimating (because each row of $Q_t$ are probabilities that sum to 1). We think it is quite reasonable to achieve learning of 2 real-valued parameters with the dataset sizes we use.
>
> However, you bring up an interesting perspective, one that we think actually demonstrates the value of our approach. Choosing a specific model is largely a practitioner's decision. They can choose their own approach and still use it with the machinery we construct. For example, we use a DNN to model $Q$ in the Continuous case, but a practitioner could use a GP or set of ODEs to achieve the same goal based on their specific use-case.
>
> > **Q2) It is interesting that both continuous and discontinuous estimations perform quite similarly on the real-world stress detection example. Does this mean that the proposed approach works best when there is large variance in noise rates?**
>
> This is an insightful observation. There are several possible explanations for this effect, one of which aligns with your suggestion. It is plausible that the performance of each method depends on the specific properties of the dataset and the practitioner’s design choices. In this case, the discontinuous method may perform well due to high variance in noise rates, where neighboring time-steps exhibit volatility. Another potential explanation is that different individuals in the dataset may follow slightly different temporal noise models, requiring each method to learn the most representative one across the entire dataset - therefore the method with the ability to capture the most variability (Discontinuous) will perform well.
>
> This observation reinforces the importance of studying temporal label noise, a largely unexplored yet important area of research. We hope that discussions like this will inspire further study in this field.
>
> > **Q3) Is there an assumption that all data instances have [an] equal number of time steps?**
>
> Not at all! Our problem setup only requires a model that takes the form $p(y_t \mid x_{1:t})$. In practice, the RNN architecture we use can handle sequences of varying lengths, even within the same dataset. However, the discontinuous approach does have a limitation: it requires all sequences to have an equal number of time steps, as a specific  $Q_t$  is defined for each time step. In contrast, the continuous approach is more flexible, as it learns  $Q(t)$ , a function of time, which can theoretically adapt to any time step $t>0$.
>
> —
>
> Thank you again for the feedback, and if there are any other questions we can answer in the discussion period, please let us know.

---

> > ### Author Response · Authors · 2024-11-25
> > **Follow-up**
> >
> > Thank you again for your valuable feedback! We've provided a detailed response to your comments. In case the OpenReview system didn't notify you, we wanted to bring it to your attention here as well.
> >
> > If you find our response unsatisfactory, please don't hesitate to share your concerns with us. We are more than happy to continue the discussion.
> >
> > Thank you for your time and consideration.

---

> > > ### Comment · Reviewer_qqUX · 2024-11-27
> > >
> > > Thank you for your detailed response. This addresses all my concerns. So, I will maintain the same score.

---

### Official Review · Reviewer_Nmrx · 2024-11-03

**Soundness:** 2
**Presentation:** 3
**Contribution:** 3
**Rating:** 6
**Confidence:** 3

**Summary:**

The paper discusses the problem of learning from time series classification task, where the label noise can vary over time. Each instance is a sequence over T time steps, where at iteration $t$ we have access to the features at time $t$, and a noisy label $\tilde y_t$ obtained from the true label $y_t$. Each of this instance is i.i.d., and the assumption is that the true label $y_t$ at time $t$ only depends on the past features $x_{1:t}$, and the noisy labels are conditionally independent to the features given by the true labels. In particular, the authors assume there exists a noise mapping $Q_t$ that describes the noise process at time $t$. They propose a novel method to learn a classifier that maps sequences $x_{1:t}$ to a label $y$. This model simultaneously approximates the noisy mapping $Q_t$ (Eq 3), that is used to approximate the true labels, which are used during training

**Strengths:**

Learning from time series data is an interesting problem. The paper addresses a challenging problem, where the label noise can change over time, and it is well-motivated. Overall, the introduction of the paper and the experimental sections are well-written and easy to follow. I found the use of the minimum-volume simplex assumption as an objective to solve the problem to be intriguing.

**Weaknesses:**

(**important**). The appendix of the paper is only a draft and it looks like it was not finished to be written. It also seems to be dissociated to the main paper.
- Appendix A is self contained and not referenced in the main paper. It is not discussed what the content of this section adds as a contribution.
- Appendix B looks dissociated to the main paper. First of all, three assumptions are introduced (they seems related to the 2 assumptions used in the main paper. In this case, why re-introduce them, and why do we use 3 assumptions rather than 2?). Most of the proofs are just a sequence of mathematical equations with text. Lines 810 to 840 are just a sequence of equations without text (maybe do a table?). The proof of Proposition 1 does not appear, which is only the theoretical result in the main paper (is Theorem 3 the proof of Proposition 1? Why does it have  different notation?).
- Section G.2 is empty. Page 25 to 47 include a sequence of a lot of figure without almost no text. I recommend the authors to only include the figures that are used to “say” something, together with a text explaination.

 The technical section is also sometimes unclear.

Notations is sometimes unclear. What is q_t in line 140? According to the notation defined in the paper $\mathcal{X} \subseteq \mathbb{R}^{d \times T}$, but the function $h_{\theta}$ that has domain $\mathcal{X}$ can also have an input matrix $d \times t$.

There is no comment on the assumptions 1 and 2 (except that they are two standard assumptions). I believe a few lines commenting on those assumptions would be helpful.

In lines 259-261, why is the Frobenius norm a convex surrogate for the volume? (A citation is also probably needed).

In Line 260, should $R_t$ be defined over $\tilde{y}$ rather than $y$?

 In lines 262-264, my understanding is that $\lambda$ is the Lagrange multiplier of the constraint of Equation~2. I would be clearer on this, since $\lambda$ does not appear in Eq 2.

**Questions:**

See also weakness.

The model does not use the noisy label in the prediction (the predictor h_{\theta} only depends on x_{1:t}).  I believe it would be interesting to include the noisy label in the prediction of the true label (i.e., if the noisy label is always accurate, can we use it for prediction? I understand this is a slightly different setting, as the goal of the paper is to learn a predictor).

The fact that we learn a noisy label structure Q_w(t) that is “continual” over time seems implicit in Eq. (3). However, it is not clear what the difference is between Eq(3) and the model discussed in “discontinuous estimation”. It seems to me that $t$ is an integer, and $Q_w(t)$ could be completely different than $Q_w(t+1)$ in principle (what is the temporal relationship discussed in line 274)

It is a bit unclear how Section 2, in particular section 2.2, is related to the proposed method in Section 3. It seems to me, that the “loss of the classifier” is embedded in the constraint of Equation 2. Lines 233-234 says that Eq(2) minimizes the forward temporal loss as in Def 2, but this loss does not actually appear on Equation 2.

Typos: 277 continuuity

---

> ### Author Response · Authors · 2024-11-22
> **Response to Reviewer Nmrx (I)**
>
> Thanks for your time and detailed feedback! We've addressed most of the questions and comments below. If there is anything else, please let us know!
>
> > **W1) “(important) …the appendix of the paper is… dissociated to the main paper”**
>
> Thank you for suggesting these revisions! We have made all suggested changes to the Appendix. In summary, we have made the following changes:
> * Streamlined the appendix by removing Appendix A (not referenced in the main article) and moving extra supplemental figures to the anonymous repository (e.g., Appendix G.2 has been removed). To clarify, Appendix A originally described a secondary, less-performant noise-tolerant sequential loss function.
> * All figures in the Appendices now have detailed captions.
> * Proposition 1 now stands out without Appendix A, further highlighting the proof of our primary theoretical result.
> * The assumptions listed in the Appendices were equivalent to the ones in the main text - we have unified the description of the assumptions to ensure that this is clear to future readers.
> * Moved definitions and quantities for proofs into a table, thank you for this suggestion!
>
>
> > **W2) Regarding notation and problem formulation:**
>
> We have revised our paper as follows to address the concerns about notation and problem formulation:
> * $q_t$ denotes a probability that is conditioned/dependent on time. We have clarified in Section 2 that this notation is to highlight this property.
> * We have elaborated on the assumptions we use to make them more intuitive for the reader. In Preliminaries (Section 2), we clarify that Assumption 1 requires that the current observation is independent of the future observations (i.e., the present does not depend on the future) and Assumption 2 specifies a feature-independent noise regime - both of which are standard and intuitive.
> * In Section 3.2, we have added a key citation on convex optimization which motivates our use of the Frobenius norm in our optimization objective.
> * We have fixed the typos in line 260 and line 277, thank you for pointing these out!
> * We have also clarified the dimensionality of the input to $h_\theta$ in Section 2.1. $h_\theta$ accepts as input $x_{1:t}$, where $t \leq T$. Thank you for pointing this out!
> * Lastly, we have improved the description of the role of the Lagrangian multiplier in Section 3.2. It is the Lagrangian multiplier used to satisfy the equality-constrained optimization problem in Eq 2.
>
> > **Q1) “If the noisy label is always accurate, can we use it for prediction?... I understand this is a slightly different setting, as the goal of the paper is to learn a predictor.”**
>
> This is an interesting idea, and is outside the scope of our problem, as you note. Our goal is to develop a time-series classifier that predicts clean labels at each time step based solely on the features. The main challenge in incorporating the noisy label into predictions is exactly as you mentioned: we cannot reliably determine when the noisy label is accurate and when it is not.  If the learner had this information then that implies they have some prior knowledge of the noise model. We do not assume this to be the case in our paper, and instead learn this noise model from data.
>
> This point is interesting, as there may be alternative strategies to learning under temporal label noise. This is a promising direction of future research and our work can inspire other such directions.
>
> > **Q2) “[Can you clarify]... the difference … between Eq(3) and the model discussed in “discontinuous estimation”. It seems to me that t is an integer, and Qw(t) could be completely different than Qw(t+1) in principle"**
>
> $Q(t)$ absolutely could be completely different from $Q(t+1)$ – this is exactly why we designed the Discontinuous method, which assumes independence between neighboring timesteps and therefore can capture this local temporal variance.
>
> So these models differ in *how they model temporal noise.* We introduced multiple methods in order to be as flexible for different practitioner needs. In short, the Discontinuous estimation learns parameters for a $Q_t$ at each time-step, treating each time-step independently. Continuous learns a unified function $Q$ with parameters $\omega$ (using a DNN) that represents a function of $t$ across all time-steps. They both use the same optimization strategy (Eq 3).
>
> To make this distinction clearer, we have added more points in Section 3.4 - where we compare and contrast Discontinuous with Continuous. We hope this clarifies, but are happy to elaborate further if needed.
>
> > **Q3) “What is the temporal relationship discussed in line 274?”**
> The temporal relationship is baked into the Continuous method, which learns a single, time-dependent noise function. The temporal relationship arises because we are explicitly learning a single, time-dependent function (i.e., temporal) to represent all time points.

---

> > ### Author Response · Authors · 2024-11-22
> > **Response to Reviewer Nmrx (II)**
> >
> > > **Q4) “How [is] Section 2, in particular section 2.2,...related to the proposed method in Section 3?”**
> >
> > The high-level structure of these two sections is as follows:
> > * Section 2.2 introduces the temporal noise-tolerant loss function we propose in our paper. Using the temporal noise function we can learn a noise-tolerant time series classifier from noisy data.
> > * Section 3 proposes a method that can identify this temporal noise function from the data while simultaneously optimizing for the loss in Section 2.2.
> >
> > For further details, our loss in Section 2.2 works by treating the noisy posterior as the matrix-vector product of a noise transition matrix and a clean-class posterior. That is, \${p}(\tilde{y}\_t \mid x\_{1:t}) = Q\_t^\top \cdot p(y \mid x\_{1:t})\$. We achieve this by minimizing the error between \$\tilde{y}\_t\$ and \$Q\_t^\top \cdot h\_\theta(x\_{1:t})\$ (see Definition 2).
> >
> > In Section 3 (Eq 2), we propose an equality-constrained optimization problem to identify the temporal noise function $Q(t)$. The equality-constraint in Eq 2 decomposes the noisy label posterior in the same exact way as Section 2.2. We can more clearly see this link in the term $R_t(\theta,\omega) = \frac{1}{n}\sum_{i=1}^{n}\ell_t(y_{t,i}, Q_\omega(t)^\top h_\theta(x_{1:t,i}))$, which is a key component of the objective we optimize in our proposed method. As you can see, $R_t(\theta,\omega)$ has the same form as the loss function we develop in Section 2.2. We can observe that this approaches zero as the equality-constraint in Eq 2 is met. The extra machinery (i.e., Augmented lagrangian optimization strategy) is to transform the constrained optimization problem in Eq 2 into a form we can minimize using standard ML optimizers.
> >
> > —
> > In the newly updated revision of the paper, we have incorporated changes to address your questions and comments. If there are any other questions we can answer in the discussion period, please let us know.

---

> > > ### Author Response · Authors · 2024-11-25
> > > **Follow-up**
> > >
> > > Thank you again for your valuable feedback! We've provided a detailed response to your comments. In case the OpenReview system didn't notify you, we wanted to bring it to your attention here as well.
> > >
> > > If you find our response unsatisfactory, please don't hesitate to share your concerns with us. We are more than happy to continue the discussion.
> > >
> > > Thank you for your time and consideration.

---

> > > > ### Comment · Reviewer_Nmrx · 2024-11-26
> > > >
> > > > I thank the authors for their detailed response and for taking action on improving the current draft. The authors improved the presentation of the appendix and technical results, and I have decided to increase my score.

---

### Official Review · Reviewer_gNk9 · 2024-11-05

**Soundness:** 3
**Presentation:** 4
**Contribution:** 3
**Rating:** 6
**Confidence:** 4

**Summary:**

The manuscript introduced temporal label noise in time series classification tasks and proposed a novel framework that are robust to it. Experiments were conducted on 4 datasets with synthetic noise injection and a real-world noisy labelled dataset, in which the method proposed in the manuscript is superior to other methods. The authors also visualized the dynamic evolution of label noise over time.

**Strengths:**

1.	The problem proposed by the authors is novel and interesting, and they evaluate it on real-world datasets.
2.	The method introduced in the manuscript is cleverly crafted and has a theoretical foundation.

**Weaknesses:**

1.	In many settings, the performance of Plug-In method is inferior to that of static methods.
2.	Only one of the five datasets in the manuscript contains both clean and noisy labels. In the remaining real-world scenarios, there is no experimental evidence to confirm that “the label noise evolves over time” actually exists.
3.	The experimental scenarios are focused on healthcare; validating the work in more diverse scenarios would broaden its applicability.

**Questions:**

1.	What is the meaning of variable d in line 126? Is that mean a multivariate time series with d variables?
2.	When performing a train-test split on the datasets, were individuals also split?

---

> ### Author Response · Authors · 2024-11-22
> **Response to Reviewer gNk9**
>
> Thanks for your time and detailed feedback! We've addressed most of the questions and comments below. If there is anything else, please let us know!
>
> >**W1) “In many settings, the performance of Plug-In method is inferior to that of static methods.”**
>
> ‘Plug-in’ is only one of three temporal methods we propose. Overall, we find that ‘Continuous’ is by-far the best temporal method, outperforming all static methods (and outperforming ‘Plug-In’). Further, ‘Plug-In’ is a temporal extension of the static ‘Anchor’ method (not ‘VolMinNet’). So the fairest comparison is ‘Plug-In’ vs ‘Anchor’, where we actually see ‘Plug-In’ consistently outperforms static ‘Anchor’. As discussed in Section 3.4, each temporal method has advantages and disadvantages, depending on the data and task. It is up to a practitioner to balance performance with the various advantages and disadvantages of each choice.
>
> We have clarified this point in the Results (Section 4.2) - thank you for pointing it out!
>
>
> > **W2) “Only one of the five datasets in the manuscript contains both clean and noisy labels. In the remaining real-world scenarios.”**
>
> A lack of data is a key issue in this research area—our work is actually a step towards fixing this issue because we show that addressing label noise improves accuracy. And by including real-world data, our work can inspire other researchers to consider how temporal label noise may affect their problems, ultimately encouraging the curation of new datasets.
>
> As the idea of temporal label noise is new, it will take time for such datasets to emerge, but our work is a crucial first step. For example, in the static label noise setting the standard dataset with both clean and noisy labels is [Clothing1M](https://paperswithcode.com/dataset/clothing1m).
> This dataset was only developed after the pace of research on static noisy labels picked up. Similarly, our work can spark progress in understanding and addressing temporal label noise.
>
> We have highlighted this point in the Conclusion (Section 6).
>
>
> > **W3) “The experimental scenarios are focused on healthcare”**
>
> Collecting time-stamped observations is indeed a common process across many fields. This paper was inspired by our own real-world work in healthcare: for instance, while collecting periodic health surveys alongside wearable device time-series data, we observed that participants often mislabeled their activities depending on what they were doing and when.
>
> That being said, if there are specific time-series datasets (with sequences of labels at each time-step) you suggest, please let us know, and we will do our best to incorporate them in a revised version during the discussion period.
>
> > **Q1) “What is the meaning of variable d … a multivariate time series with d variables?”**
>
> Yes, that is correct! We have updated the Preliminaries (Section 2) to clarify that our inputs are multivariate time series with d variables.
>
> > **Q2) “When performing a train-test split on the datasets, were individuals also split?”**
>
> Splitting strategies depend on the dataset. For example, in the real-world stress detection demonstration, the training and testing splits did not share individuals, as there is one time series per individual. For other datasets, such as ‘moving’ and ‘senior’, we used the given train-test splits.
>
> We have added this information to Dataset Details (Appendix E.1) so it is clear for future readers.

---

> > ### Author Response · Authors · 2024-11-25
> > **Follow-up**
> >
> > Thank you again for your valuable feedback! We've provided a detailed response to your comments. In case the OpenReview system didn't notify you, we wanted to bring it to your attention here as well.
> >
> > If you find our response unsatisfactory, please don't hesitate to share your concerns with us. We are more than happy to continue the discussion.
> >
> > Thank you for your time and consideration.

---

### Author Response · Authors · 2024-11-22
**Common Response**

We thank all reviewers for their time and feedback!

We are pleased that reviewers recognized the **“novel”**, **“interesting”**, and **“challenging”** [gNk9, EoQ6, Nmrx] problem of temporal label noise introduced in our work.  Reviewers appreciated our **“cleverly crafted”** [gNk9] framework, which leverages a **“theoretical foundation”**[gNk9] and **“clear and concise methodology”**[qqUX]. Overall, reviewers found our paper **“well-organized and clearly written”**[EoQ6] with experiments **“demonstrat[ing] practical utility”**[qqUX] in real-world datasets. Additionally, reviewers noted the **“valuable empirical results”**[qqUX] that illustrate our approach’s robustness across various temporal noise settings and datasets.

We have responded to each reviewer’s comments individually and have also uploaded an updated version of the paper incorporating feedback from all reviewers. A brief summary of the changes is as follows:
* We have made changes to the paper based on reviewer feedback for clarity. In particular:
  * Discussed the role of Assumptions 1 and 2 and their intuitive meaning
  * Sign-posting in Section 3 to motivate the procedures we construct to learn the temporal noise function from noisy data
  * Further clarification on the difference between Continuous and Discontinuous methods in Section 3.4
* We have polished the Appendix based on reviewer Nmrx’s suggestions. This includes moving extra supplemental figures not referenced in the text to our anonymous repo, and making the notation/proofs easier to follow.
* We have added a Limitations and Future Work section, which addresses questions from reviewers and also outlines future research directions that our work can inspire.

Overall, we believe incorporating this feedback has improved the paper. We thank the reviewers for helping us further refine the paper and look forward to answering any remaining questions over the coming days.

–

If you feel we have not sufficiently addressed your concerns to motivate increasing your score, we would love to hear from you further on what points of concern remain and how we can improve the work in your eyes. Thank you again!

---

### Meta-Review · Area_Chair_RtMs · 2024-12-14

**Metareview:**

A fairly good paper that should be accepted for publication at ICLR. I hope this paper can advance the research area label-noise learning.

My only comment is about related work. The standard non-temporal label noise can be regarded as a type of distribution shift, where the test distribution is clean and the training distribution changes to some noisy version. Then, the temporal label noise is actually a type of continuous distribution shift, where the change from test to training is also time-dependent. Even though continuous distribution shift focused on continuous covariate shift rather than continuous class-posterior shift (i.e., label noise), it is still related to temporal label noise as a bigger topic covering the problem under consideration. However, the term distribution shift didn't appear at all. I hope you can acknowledge that your temporal label noise is a special case of continuous distribution shift, so that you have also contributed to distribution shift research in addition to label-noise research.

**Additional Comments On Reviewer Discussion:**

The rebuttal addressed most of the concerns from the reviewers.

---

### Decision · Program_Chairs · 2025-01-22

Accept (Poster)